# CausalTAD: Injecting Causal Knowledge into Large Language Models for Tabular Anomaly Detection

## Abstract

Detecting anomalies in tabular data is critical for many real-world applications, such as credit card fraud detection. With the rapid advancements in large language models (LLMs), state-of-the-art performance in tabular anomaly detection has been achieved by converting tabular data into text and fine-tuning LLMs. However, these methods randomly order columns during conversion, without considering the causal relationships between them, which is crucial for accurately detecting anomalies. In this paper, we present CausalTAD, a method that injects causal knowledge into LLMs for tabular anomaly detection. We first identify the causal relationships between columns and re-order them to align with these causal relationships. This reordering can be modeled as a linear ordering problem. Since each column contributes differently to the causal relationships, we further propose a reweighting strategy to assign different weights to different columns to enhance this effect. Experiments across more than 30 datasets demonstrate that our method consistently outperforms the current state-of-the-art methods.

## 1. Introduction

Tabular data plays a central role in the era of big data, powering applications from financial analysis to healthcare diagnostics (Jiang et al., 2025). Among these data, a large portion includes free-text columns, such as user reviews, product descriptions, or clinical notes (Arazi et al., 2025). These free-text columns are important for detecting anomalies among tabular data. For example, in user reviews, unusual sentiment or specific word patterns might signal fraudulent behavior or product defects. However, on the other hand, these free-text columns also present challenges for

anomaly detection. Specifically, existing anomaly detection methods typically rely on techniques such as Word2Vec to convert free-text columns into vectors. Such a strategy often struggles to capture the nuanced meaning and context of the text, leading to potential limitations in identifying subtle anomalies or understanding domain-specific language.

Recent advances in large language models (LLMs) offer a promising alternative to address the issue of free texts in anomaly detection. The SOTA method, AnoLLM (Tsai et al., 2025), serializes tabular data, including the free texts, into texts with random orders and fine-tunes an LLM to model the generative probability distribution. Then, instead of using the generative probability of the final column as the anomaly score, which has been shown to result in lower performance, the anomaly score of a sample is computed based on the average of the generative probabilities for each column. These probabilities are conditioned on the columns that precede it in the serialized texts.

While achieving SOTA performance, AnoLLM does not consider the causal relationships between columns in the serialization, which affect the detection performance. For example, Figure 1 illustrates how column ordering can influence LLM-based anomaly detection in tabular data. In Figure 1(a), the order of the columns is random, so the "education" and "technical skill" columns may come before "job descriptions." As a result, their probabilities conditioned on the "salary" column are low, since the required education and skills are relatively low, while the salary is very high. This makes the sample appear anomalous. However, the sample is actually normal because the job is in extremely hazardous conditions, which justifies the high salary. As shown in Figure 1(b), when the columns are ordered according to causal relationships, the conditional probabilities of these columns are high, leading to the correct prediction that the sample is normal.

These observations hint that incorporating causal relationships into tabular anomaly detection could enhance performance. However, despite there are many causal discovery methods in the literature (Liu et al., 2024a), integrating them into the tabular anomaly detection process still faces two main challenges. **Challenge 1: Column ordering.** The causal relationships identified by the causal discovery meth-

---

[1]Anonymous Institution, Anonymous City, Anonymous Region, Anonymous Country. Correspondence to: Anonymous Author <anon.email@domain.com>.

Preliminary work. Under review by the International Conference on Machine Learning (ICML). Do not distribute.

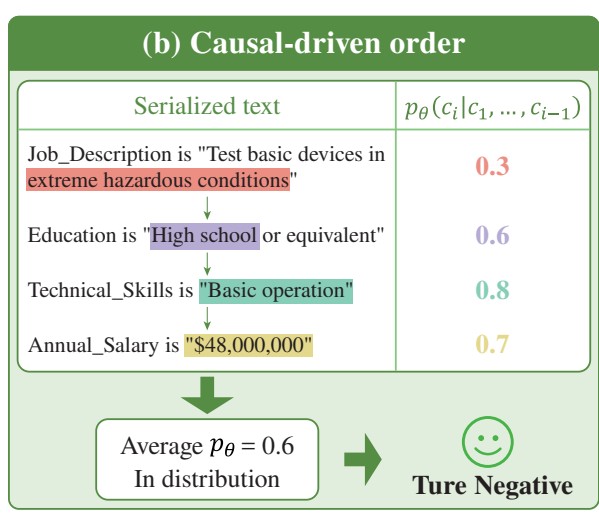

*Figure 1.* An illustrative example highlighting the impact of column ordering on LLM-based tabular anomaly detection.

ods usually form graphs, but the columns must be serialized to be input to LLMs. Therefore, it is necessary to find column orders that can most preserve the causal relationships, which remains unclear. **Challenge 2: Column reweighting.** Not all columns contribute equally to the causal relationships; some have a stronger influence, while others have a weaker one. Columns with weak causal relationships have a minimal impact on the anomaly detection results. Therefore, it is important to account for these varying effects during the detection process.

To address these challenges, we present CausalTAD, a method that injects causal knowledge into LLMs for tabular anomaly detection. It consists of two main modules: a causal-driven column ordering method and a causal-aware reweighting method. The causal-driven column ordering method extracts latent factors from the tabular data and construct a causal graph among the factors. Factors are high-level concepts and are described by one or more columns. For example, a factor "compensation" is described by the "salary" and "benefits" columns. Based on the causal graph, this method reorders the columns, which can be formulated as a **Linear Ordering Problem (LOP)**. The causal-aware reweighting method assigns different weights to different columns to enhance the affects of columns with strong causal relationships. Experiments across more than 30 datasets demonstrate that our method consistently outperforms the current state-of-the-art methods.

In summary, our contributions are as follows.

- **A causal-driven column ordering method** to reorder columns in tabular data aligned with causal relationships.
- **A causal-aware reweighting method** to assign different weights to different columns to enhance the effects of columns with strong causal relationships.

- **Empirical experiments** across more than 30 datasets demonstrate the effectiveness of our method.

## 2. Related Work

**LLM-based Anomaly Detection.** Recent advances in large language models (LLMs) have prompted their application to anomaly detection across various domains. For time series, LLMAD(Liu et al., 2024b) employs in-context learning, while A2P(Park et al., 2025) introduces anomaly prompting. In vision, VadCLIP(Wu et al., 2023) and AnomalyGPT(Gu et al., 2023) leverage vision-language models, with AnomalyRuler(Yang et al., 2024) applying rule-based reasoning for video anomaly detection. For log data, LogFiT(Almodovar et al., 2024) and subsequent works(Lim et al., 2025; Song et al., 2025) fine-tune language models to capture linguistic patterns. For tabular data, AnoLLM(Tsai et al., 2025) pioneers serializing tables into text for LLM-based detection, preserving mixed-type features and textual information. Li et al. (2024) demonstrate pre-trained LLMs as zero-shot batch-level detectors, while ReTabAD (Yoon et al., 2025) incorporates semantic context through textual metadata. Despite these advances, existing methods randomly order columns during serialization without considering causal relationships crucial for accurate detection.

**Traditional Tabular Anomaly Detection.** Classical methods include density-based approaches(Breunig et al., 2000), distance metrics(Ramaswamy et al., 2000), isolation mechanisms(Liu et al., 2008), and one-class classification(Schölkopf et al., 1999), with k-nearest neighbors remaining a strong baseline(Livernoche et al., 2023). Deep learning methods can be categorized into margin-based approaches, such as DeepSVDD (Ruff et al., 2018) and DROCC (Goyal et al., 2020), which map normal data into

minimal-volume hyperspaces, and self-supervised methods, including NeuTral (Qiu et al., 2021), REPEN (Pang et al., 2018), SLAD (Xu et al., 2023), and ICL (Shenkar & Wolf, 2022), which leverage contrastive objectives to model data normality. Recent advances have introduced noise evaluation (Dai et al., 2024), partition-based contrastive learning (Li et al., 2025), and retrieval-augmented reconstruction (Thimonier et al., 2024) to enhance detection performance. However, these methods struggle with raw text processing (Taha & Hadi, 2019) and exhibit limited capability in handling mixed-type datasets containing free-text columns, as they lack robust semantic modeling for textual features.

## 3. Problem Formulation

Following AnoLLM (Tsai et al., 2025), we adopt the standard contamination-free, unsupervised anomaly detection setting. We are given a training set $\mathcal{X} = \{X_1, \ldots, X_m\}$ containing only normal samples. Each sample $X = [x_1, \ldots, x_d]$ consists of $d$ features (columns), where each feature $x_j$ can be numerical, categorical, or free text. We denote the set of column names as $C = [c_1, \ldots, c_d]$, where each column name $c_j$ is represented as a natural language text sequence (e.g., "age", "vehicle description"). Our goal is to learn a detector $\mathcal{D} : \mathcal{X} \to \mathbb{R}$ that assigns higher anomaly scores to test samples deviating from the normal distribution. At test time, we evaluate $\mathcal{D}$ on a labeled test set $D' = \{(X_1', Y_1'), \ldots, (X_{n_{\text{test}}}', Y_{n_{\text{test}}}')\}$, where $Y_i' \in \{0, 1\}$ with $Y_i' = 1$ indicating an anomaly.

## 4. Method

As shown in Figure 2, CausalTAD consists of two main modules: a causal-driven column ordering method to reorder columns aligned with causal relationships, and a causal-aware column reweighting method to assign different weights to different columns to enhance the effects of columns with strong causal relationships.

### 4.1. Causal-Driven Column Ordering

To reorder the columns of tabular data in a way that reflects causal relationships, we first discover the causal relationships (Sec. 4.1.1) and reorder columns to align with these relationships (Sec. 4.1.2).

#### 4.1.1. CAUSAL DISCOVERY

We do not extract causal relationships directly among columns for two reasons. First, columns are often low-level, non-causal variables (e.g., medicine ID), whereas meaningful causal structure typically exists among higher-level latent factors (Deng et al., 2025). Second, the presence of free-text columns makes it difficult to reliably determine

causal relationships at the column level. Therefore, we utilize a causal discovery method adapted from the latest COAT framework (Liu et al., 2024a) to extract high-level factors and causal relationships among these factors.

COAT is originally designed for unstructured data (e.g., text) and operates through iterative factor proposal and causal discovery. We extend it to handle mixed tabular data by serializing each sample into natural language text. Specifically, for a sample $X = [x_1, \ldots, x_d]$ with column names $C = [c_1, \ldots, c_d]$, we construct the text representation as "$c_1$ is $x_1$, $c_2$ is $x_2$, $\ldots$, $c_d$ is $x_d$". This serialization preserves all information while enabling COAT to process structured data alongside free-text columns, which traditional causal discovery algorithms cannot handle directly.

The adapted framework of COAT proceeds as follows. First, the serialized training samples are input to LLMs to extract a set of factors $F = \{f_1, \ldots, f_k\}$ (the prompts can be found in Appendix D). Each factor $f_i$ represents a high-level concept with discrete possible values and annotation criteria. Take the factor "Compensation" in Fig. 2 as an example, this factor can be represented by the "salary" and "benefits" columns. Its possible values range from 0 to 2, representing basic, standard, and premium packages, respectively. The annotation criteria specify how to combine column values to assign factor values; for example, if the "salary" column is less than 20000 and the "benefit" column is "low", then the factor value is 0. Then, based on the annotation criteria, LLMs automatically annotate the factor values of each sample, resulting in a factor value matrix $\mathcal{F}^{m \times k}$. We can also obtain the mapping from columns to factors $\mathcal{M}^{k \times d} : F \to 2^C$. $\mathcal{M}_{i,j} = 1$ indicates that the $i$-th factor is described by the $j$-th columns; $\mathcal{M}_{i,j} = 0$ otherwise. This mapping is many-to-many: a single factor may be derived from multiple columns, and one column may contribute to multiple factors.

With the factor value matrix $\mathcal{F}^{m \times k}$, we can apply off-the-shelf causal discovery algorithms to identify the factor-level causal graph $G_F = (F, E_F, W)$. $E_F$ denotes the set of directed causal edges and $W : E_F \to \mathbb{R}$ denotes the weights of the edges. In theory, any causal discovery algorithm can be applied here. In our experiments, we adopt three types of widely-used causal discovery algorithms, including PC Spirtes & Glymour (1991), LiNGAM Shimizu et al. (2006), and FCI Spirtes et al. (2000), and compared their performance (see Sec. 5.3 for detailed results.)

#### 4.1.2. CAUSAL-DRIVEN LINEAR ORDERING

**Projecting Factor-Level Causality to Columns.** Our core principle for column ordering is that if column $c_i$ has a strong causal influence on column $c_j$, then $c_i$ should precede $c_j$ in the serialized sequence. However, the causal graph $G_F$ describes relationships among high-level factors, not

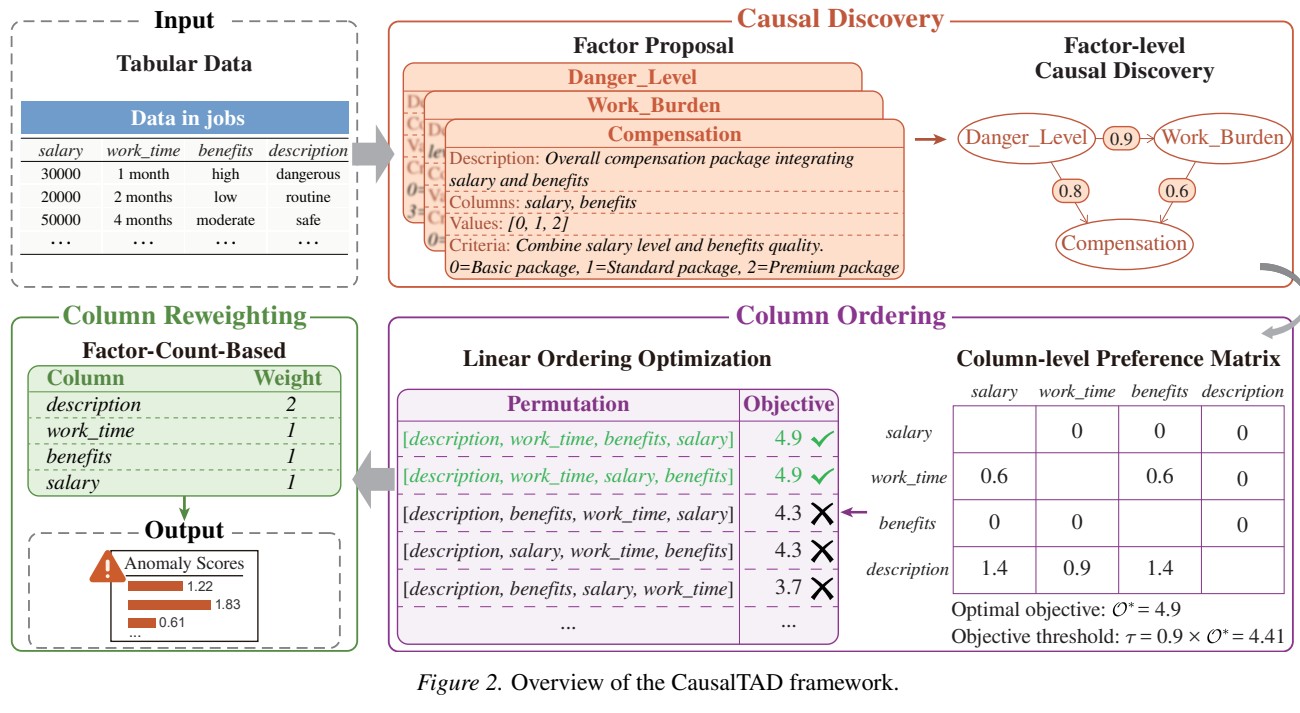

*Figure 2.* Overview of the CausalTAD framework.

directly between columns. To bridge this gap, we project the factor-level causal relationships onto columns using the factor-to-column mapping $M(\cdot)$. For any two columns $c_i$ and $c_j$, we compute a causal preference strength $w(c_i \rightarrow c_j)$ by aggregating causal strengths across all pairs of factors that involve these two columns:

$$w(c_i \rightarrow c_j) = \sum_{\substack{f_a \in M^{-1}(c_i) \\ f_b \in M^{-1}(c_j)}} |W_{f_a \rightarrow f_b}|, \quad (1)$$

Here, $W_{f_a \rightarrow f_b}$ is the causal edge weight of $f_a \rightarrow f_b$ in $G_F$, and $M^{-1}(c)$ denotes the inverse mapping set of factors that involve column $c$. The sign of $W_{f_a \rightarrow f_b}$ indicates whether the causal effect is promoting (positive) or inhibiting (negative). For the purpose of ordering, however, we are concerned only with the strength of the dependency, not its direction. Therefore, we take the absolute value $|W_{f_a \rightarrow f_b}|$ to quantify the magnitude of the causal influence. Intuitively, a column may participate in several factors; each causal edge between a factor involving $c_i$ and a factor involving $c_j$ provides evidence for placing $c_i$ before $c_j$. Summing $|W_{f_a \rightarrow f_b}|$ across all such factor pairs aggregates these contributions into a single column-level preference strength. This projection yields a column-level preference matrix $\mathcal{W} \in \mathbb{R}^{|C| \times |C|}$ with entries $\mathcal{W}_{ij} = w(c_i \rightarrow c_j)$.

**Formulation as a Linear Ordering Problem.** The preference matrix $\mathcal{W}$ does not guarantee to be acyclic: for a pair of columns $c_i$ and $c_j$, it is possible that both $\mathcal{W}_{ij} > 0$ and $\mathcal{W}_{ji} > 0$, reflecting the mixed or entangled causal associations commonly observed in real data. Even when

the factor-level graph $G_F$ is acyclic, the projection onto columns can introduce cycles. For example, if factors form a chain $f_1 \rightarrow f_2 \rightarrow f_3$ and the columns map as $c_1 \in M(f_1)$, $c_2 \in M(f_2)$, while $c_3 \in (M(f_1) \cap M(f_3))$, the projection can produce $w(c_1 \rightarrow c_2) > 0$, $w(c_2 \rightarrow c_3) > 0$, and $w(c_3 \rightarrow c_1) > 0$, yielding a column-level cycle. More generally, $S$ is neither guaranteed to be acyclic nor transitive: bidirectional dependencies at the factor level create conflicting precedence constraints, and indirect causal paths can produce complex dependency patterns. Therefore, a permutation that satisfies all preferences may not exist; instead, we seek a permutation that maximizes the total weight of satisfied preferences, which naturally leads to the Linear Ordering Problem (LOP).

Formally, given the column set $C$ and the preference matrix $\mathcal{W}$, the LOP seeks a linear permutation $\pi : C \rightarrow \{1, \ldots, |C|\}$ that maximizes the sum of weights for all satisfied precedence constraints:

$$\max_{\pi} \sum_{c_i \in C} \sum_{c_j \in C} w(c_i \rightarrow c_j) \cdot \mathbb{I}\left[\pi(c_i) < \pi(c_j)\right], \quad (2)$$

where $\pi(c)$ denotes the position of column $c$ in the permutation (ranging from 1 to $|C|$), and $\mathbb{I}[\cdot]$ is the indicator function, equal to 1 when its argument is true and 0 otherwise. This combinatorial optimization problem balances competing precedence constraints to find orderings that maximize total satisfaction.

**Optimization Algorithm.** Optimizing LOP is NP-hard (Gr"otschel et al., 1984). However, we adopt the

enumeration strategy to solve it for two reasons. First, the number of columns is typically small (in the tens). An enumeration strategy can find the optimal solution $\mathcal{O}^*$ within an acceptable time, which is negligible compared to the subsequent LLM fine-tuning process. Second, as described above, after the linear ordering, it is not guaranteed that all causal relationships are preserved. The objective of LOP, which maximizes the sum of weights for all satisfied precedence constraints, does not necessarily ensure the optimal preservation of causal relationships. Therefore, we consider solutions within 90% of the optimal solution as feasible, and average over multiple feasible solutions to reduce the effect of noise. Therefore, we utilize an enumeration-based method to solve Equation (2).

Our enumeration-based method for LOP proceeds in two main phases. First, we solve the LOP via an integer programming solver (e.g., CP-SAT Solver (Perron & Didier)) to obtain the optimal objective value $\mathcal{O}^*$. Then, we enumerate all column permutations whose objective values exceed a threshold $\tau = 0.9 \times \mathcal{O}^*$, and select the top-$K$ permutations for downstream use. While LOP is NP-hard and approximate solvers exist, the column count in our tabular datasets is small enough (typically tens of columns) that exact enumeration remains computationally feasible.

**Computational Time Analysis.** The time cost of our ordering algorithm mainly comes from three parts. 1) **Weight matrix computation:** For each edge $(f_u \rightarrow f_v)$ in the factor causal graph, we need to traverse the Cartesian product of its associated columns. Thus, this step has a complexity of $O(|E_F| \cdot d_{\max}^2)$, where $d_{\max} = \max_{f \in F} |M(f)|$ is the maximum number of columns associated with a single factor, which is typically small in practice. 2) **Integer programming solving:** Solving the LOP defined by Eq. (2) is NP-hard. However, for the typical scale of tabular data (tens of columns), modern integer programming solvers can find the optimal objective $\mathcal{O}^*$ within an acceptable time. 3) **Solution enumeration:** Enumerating all feasible solutions is exponential in the worst case. However, by setting a relative threshold $\tau$, we limit the solution space for enumeration. In practice, the enumeration finishes quickly.

### 4.1.3. TRAINING STRATEGY

Equipped with the pre-computed top-$K$ causal-driven column orderings, we adopt the same fine-tuning protocol as AnoLLM (Tsai et al., 2025), with the sole modification that we serialize each training sample using our causal-driven orderings rather than a random one. This causal-driven ordering helps the model capture inter-column dependencies, thereby improving next-column prediction under the autoregressive objective. During training, we randomly select one ordering from the $K$ pre-computed orderings for each training sample and serialize the corresponding data accordingly.

---

**Algorithm 1** Causal-Driven Column Ordering

**Require:** Column names $C$, Factor causal graph $G_F = (F, E_F, W)$, factor-to-column mapping $M(\cdot)$, number of selected orderings $K$

**Ensure:** Set $\varphi = \{(\pi_1, s_1), (\pi_2, s_2), \dots, (\pi_K, s_K)\}$ of top-$K$ column orderings with scores

1: Initialize column-level weight matrix $\mathcal{W} \in \mathbb{R}^{|C| \times |C|}$ with all elements 0
2: **for** each $(f_u \rightarrow f_v) \in E_F$ **do**
3:    $C_u \leftarrow M(f_u), C_v \leftarrow M(f_v)$
4:    **for** each $(c_i, c_j) \in C_u \times C_v$ with $c_i \neq c_j$ **do**
5:       $\mathcal{W}_{i,j} \leftarrow \mathcal{W}_{i,j} + |W_{f_u \rightarrow f_v}|$
6:    **end for**
7: **end for**
8: Initialize position variables $\text{pos}[c] \in \{1, \dots, |C|\}$ for all $c \in C$
9: Add constraint: AllDifferent($\{\text{pos}[c] : c \in C\}$)
10: objective $\leftarrow \sum_{c_i} \sum_{c_j} \mathcal{W}_{i,j} \cdot \mathbb{I}[\text{pos}[c_i]] < \text{pos}[c_j]]$ $\{c_i, c_j \in C \text{ and } c_i \neq c_j\}$
11: $\mathcal{O}^* \leftarrow \text{Solve(maximize objective)}$ {Optimal IP objective value}
12: $\tau \leftarrow 0.9 \times \mathcal{O}^*$ {Acceptance threshold}
13: Solutions $\leftarrow$ EnumerateAllSolutions(objective $\geq \tau$)
14: $\phi \leftarrow \{(\pi, \text{EvaluateScore}(\pi, \mathcal{W})) : \pi \in \text{Solutions}\}$
15: $\varphi \leftarrow \text{TopK}(\phi, K)$ {Select top-$K$ orderings by score}
16: **return** $\varphi$

---

Let $\mathcal{T} = \{T_\pi(X_i) \mid \pi \in \{\pi_1, \dots, \pi_K\}, \ i \in \{1, \dots, m\}\}$ be the set of serialized textual sequences. For each $r \in \mathcal{T}$, let $g(r) = (t_1, \dots, t_{l(r)})$ denote its tokenized sequence. We fine-tune the LLM using the standard causal language modeling loss, which is precisely the conventional autoregressive cross-entropy loss:

$$\mathcal{L}_\theta = \mathbb{E}_{r \in \mathcal{T}} \left[ -\sum_{k=1}^{l(r)} \log p_\theta(t_k \mid t_1, \dots, t_{k-1}) \right], \quad (3)$$

where $\theta$ denotes the model parameters, $p_\theta$ is the LLM's probability distribution, and $l(r)$ is the token length of $r$.

### 4.2. Causal-Aware Column Reweighting

Beyond column ordering, different columns contribute unequally to the causal structure, so they should not be treated identically when scoring anomalies. To address this, we propose a causal-aware reweighting method based on the following insight: **the number of factors a column maps to reflects its contribution to the causal structure**. Intuitively, if a column contributes to more factors, it contributes to more causal relations in the causal graph; therefore, its influence should be emphasized when calculating anomaly scores. We only need relative weights here, using factor counts as a proxy instead of quantifying precise causal

*Table 1.* Dataset statistics for the six datasets from the mixed-type benchmark. # text, # num, and # category denote the numbers of textual, numerical, and categorical features, respectively.

| Datasets | # Samples | # text | # num | # category | # anomaly (%) |
|---|---|---|---|---|---|
| Fake job posts (Grover et al., 2023) | 17,880 | 5 | 3 | 8 | 866 (4.84%) |
| Fraud ecommerce (Grover et al., 2023) | 151,112 | 0 | 1 | 6 | 14,151 (9.36%) |
| Lymphography (Rayana, 2016) | 148 | 0 | 3 | 15 | 6 (4.05%) |
| Seismic (Rayana, 2016) | 2,584 | 0 | 14 | 4 | 170 (6.58%) |
| Vehicle insurance (Kaggle) | 15,420 | 0 | 8 | 24 | 923 (5.99%) |
| 20 newsgroup (Mitchell, 1997) | 11,905 | 1 | 0 | 0 | 591 (4.96%) |

strength. Specifically, we define the causal contribution weight of column $c_j$ as:

$$\alpha_j = |M^{-1}(c_j)|, \qquad (4)$$

Given a sample $X$ and a specific column ordering $\pi_z$ from our pre-computed set, we first compute the column-wise negative log-likelihood. For column $c_j$, its score $\ell_j(X, \pi_z)$ is the average token-level prediction loss within that column:

$$\ell_j(X, \pi_z) = \frac{1}{l(c_j)} \sum_{k=1}^{l(c_j)} - \log p_\theta(t_k \mid t_1, \ldots, t_{k-1}), \quad (5)$$

where $p_\theta$ is the LLM's probability distribution. The conditioning context $t_1, \ldots, t_{k-1}$ includes all tokens from columns preceding $c_j$ in ordering $\pi_z$ and the preceding tokens within the same column. For numerical or categorical columns, $l(c_j)$ is typically 1, making the average equivalent to the single-token loss.

Finally, the anomaly score for sample $X$ is obtained by averaging the weighted column scores across all $K$ causal-driven orderings:

$$\text{Score}(X) = \frac{1}{K} \sum_{t=1}^{K} \sum_{j=1}^{d} \alpha_j \cdot \ell_j(X, \pi_z), \qquad (6)$$

This formulation ensures that columns with stronger connections to the causal factor graph exert a larger influence on the anomaly score. Combined with the causal-driven orderings, this weighting scheme aligns the model's scoring behavior with the causal structure of the data.

## 5. Experiments

### 5.1. Experimental Settings

**Datasets.** Since most popular anomaly detection benchmarks are numerical or categorical, we follow AnoLLM (Tsai et al., 2025) and use the same **six mixed-type datasets** sourced from the ODDS library (Rayana, 2016) and Kaggle. Among them, Fake job posts and 20 newsgroup contain free-text columns. Table 1 summarizes the six mixed-type datasets used in our main experiments.

Detailed dataset statistics are provided in the Appendix B. To further demonstrate the ability to handle purely numerical data, we also evaluate on **30 more numerical datasets** from ODDS; their details are reported in the Appendix C.

**Baselines.** We compare CausalTAD against a diverse set of baseline tabular anomaly detection methods spanning classical machine learning methods and deep learning methods. Classical methods include Isolation Forest (Liu et al., 2008), Principal Component Analysis (PCA), k-Nearest Neighbors (KNN) (Ramaswamy et al., 2000), and ECOD (Li et al., 2022). Deep learning methods include DeepSVDD (Ruff et al., 2018), RCA (Rakhi et al., 2024), SLAD (Xu et al., 2023), GOAD, NeuTral (Qiu et al., 2021), ICL (Shenkar & Wolf, 2022), DTE, and REPEN (Pang et al., 2018). We also compare against the state-of-the-art LLM-based method **AnoLLM** (Tsai et al., 2025).

**Evaluation Metrics.** Following prior works (Tsai et al., 2025), we conduct experiments in an uncontaminated, unsupervised setting. The training set consists of a random sample of 50% of the normal examples, while the test set includes the remaining normal examples along with all anomalies. We adopt two widely used anomaly detection measures, the Receiver Operating Characteristic Curve (AUC-ROC) and the F1 score. Since both of them lead to similar experimental conclusions, we only report the AUC-ROC scores in this section and leave the F1 scores in the Appendix C.

**Implementation Details.** To enable a fair comparison with AnoLLM, we also adopt SmolLM-135M and SmolLM-360M (Allal et al., 2024) as the backbone LLMs. We also compared other LLMs in the experiments. The training process uses a learning rate of 5e-3 with a maximum of 18,000 training steps. All experiments are conducted on 4×NVIDIA RTX 4090 GPUs with 24GB memory each. For causal discovery, we employ the COAT framework (Liu et al., 2024a) for factor extraction, utilizing GPT-4 for factor proposal and DeepSeek-R1-Distill-Qwen-32B for factor annotation. We use the same set of hyperparameters across all main datasets to ensure fair comparison and demonstrate the robustness of our method. For baseline methods, we either directly cite results from the original AnoLLM paper when available, or

*Table 2.* AUC-ROC scores for all methods on the six mixed-type tabular datasets. **Bold** indicates the best performance. Results marked with † are cited from the original AnoLLM paper (Tsai et al., 2025).

| Methods \ Datasets | Fake job posts | Fraud ecommerce | Lympho­graphy | Seismic | Vehicle insurance | 20news groups | Average |
|---|---|---|---|---|---|---|---|
| *Classical methods* | | | | | | | |
| IForest† | 0.755 | 0.501 | 0.673 | 0.692 | 0.496 | 0.623 | 0.623 |
| PCA† | 0.724 | 0.647 | 0.826 | 0.692 | 0.509 | 0.623 | 0.670 |
| KNN† | 0.636 | **1** | 0.860 | 0.738 | 0.524 | 0.605 | 0.727 |
| ECOD† | 0.512 | 0.755 | 0.830 | 0.692 | 0.509 | 0.620 | 0.653 |
| *Deep learning based methods* | | | | | | | |
| DeepSVDD† | 0.561 | **1** | 0.899 | 0.713 | 0.505 | 0.597 | 0.713 |
| RCA† | 0.629 | **1** | 0.979 | 0.727 | 0.531 | 0.546 | 0.725 |
| SLAD† | 0.603 | 0.998 | 0.964 | 0.714 | 0.556 | 0.640 | 0.746 |
| GOAD† | 0.566 | 0.998 | 0.817 | 0.717 | 0.512 | 0.630 | 0.707 |
| NeuTral† | 0.548 | **1** | 0.847 | 0.681 | 0.507 | 0.658 | 0.707 |
| ICL† | 0.699 | **1** | 0.827 | 0.719 | 0.501 | 0.671 | 0.736 |
| DTE† | 0.548 | **1** | 0.909 | 0.714 | 0.512 | 0.600 | 0.714 |
| REPEN† | 0.653 | **1** | 0.808 | 0.724 | 0.513 | 0.574 | 0.712 |
| *AnoLLM* | | | | | | | |
| SmolLM-135M† | 0.800 | **1** | 0.968 | 0.712 | 0.569 | **0.766** | 0.803 |
| SmolLM-360M† | 0.814 | **1** | 0.995 | 0.746 | 0.555 | 0.752 | 0.810 |
| *CausalTAD (Ours)* | | | | | | | |
| SmolLM-135M | **0.873** | **1** | **1** | 0.783 | 0.581 | **0.766** | **0.834** |
| SmolLM-360M | 0.868 | **1** | **1** | **0.791** | **0.589** | 0.752 | 0.833 |

reproduce them following the exact settings described in their publications.

## 5.2. Main Results

The individual AUC-ROC scores for all baselines and our method across the six mixed-type datasets are presented in Table 2. The averaged AUC-ROC scores on the 30 numerical datasets are shown in Figure 3, with detailed results provided in Appendix C due to space limitations. From the table and figure, we draw the following observations.

(1) **CausalTAD achieves the best performance among all the datasets.** On the six mixed-type datasets, CausalTAD improves over AnoLLM on both model scales, achieving average AUC-ROC scores of 0.834 and 0.833 with SmolLM-135M and SmolLM-360M, respectively, surpassing all baseline methods. Note that the 20 newsgroups dataset contains only a single column; consequently, neither column ordering nor reweighting takes effect, making our method identical to AnoLLM on this dataset, which is a rare special case. On the 30 numerical datasets, CausalTAD still achieves the best average AUC-ROC scores. This demonstrates the effectiveness of injecting causal knowledge into LLMs for tabular anomaly detection.

(2) **On datasets with rich textual information and complex column interactions, such as Fake job posts,**

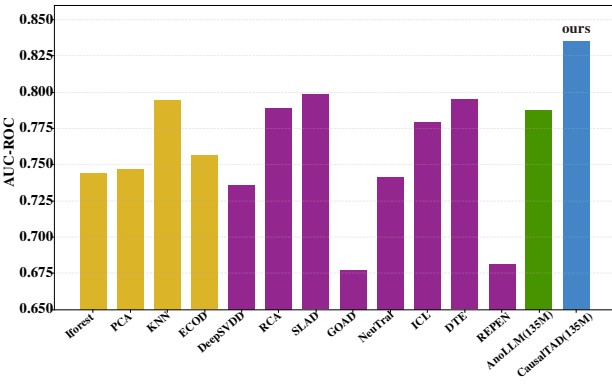

*Figure 3.* Average AUC-ROC scores of 30 numerical datasets for different methods.

**CausalTAD achieves the most significant improvements.** Specifically, with SmolLM-135M, CausalTAD improves AUC-ROC from 0.800 (AnoLLM) to 0.873—a relative improvement of 9.1%. This highlights the importance of aligning column ordering with causal relationships when processing mixed-type data with substantial textual content. For datasets with predominantly numerical features, such as Seismic and Vehicle insurance, our method still maintains clear advantages over AnoLLM. On the Seismic dataset, CausalTAD (SmolLM-360M) improves from 0.746 to 0.791. Similarly, on Vehicle insurance, it improves from 0.555 to

*Table 3.* Ablation study on different column ordering and reweighting strategies on the Fake job posts dataset. Results show AUC-ROC scores.

| Weighting \ ordering | Random | PC | LiNGAM | FCI |
|---|---|---|---|---|
| Factor-count weighting | 0.832 | 0.870 | 0.873 | 0.872 |
| Original weighting | 0.800 | 0.844 | 0.842 | 0.843 |

*Table 4.* Ablation study on different model sizes. Results show AUC-ROC scores on the Seismic dataset.

| Model Size | AUC-ROC |
|---|---|
| SmolLM-135M | 0.783 |
| SmolLM-360M | 0.791 |
| SmolLM-1.7B | 0.785 |

0.589. These results indicate that causal-driven column ordering and reweighting benefit anomaly detection across diverse feature types, not just text-rich datasets.

(3) **CausalTAD is more robust than AnoLLM.** CausalTAD consistently outperforms all baseline methods on all datasets. In contrast, AnoLLM shows unstable performance gains relative to classical/deep-learning-based baselines. For example, on the Seismic dataset, AnoLLM (SmolLM-135M) achieves an AUC-ROC of 0.712, which is lower than classical methods, such as KNN (0.738). This is because AnoLLM randomly reorders the columns, which may disrupt the causal relationships among them. By preserving the causal relationships, our method achieves consistently performance improvement. For example, CausalTAD achieves 0.783 (SmolLM-135M) and 0.791 (SmolLM-360M) on Seismic, clearly outperforming both AnoLLM and classical/deep-learning-based baselines.

### 5.3. Ablation Study

To understand the contribution of different components in CausalTAD, we conduct comprehensive ablation studies on the Fake job posts dataset, which contains rich textual information and exhibits complex causal relationships.

**Effect of Column Ordering and Reweighting Strategies.** Table 3 presents the ablation results comparing different column ordering strategies and weighting schemes. We compare different column orderings based on three causal discovery algorithms: (1) PC-based ordering, (2) LiNGAM-based ordering, and (3) FCI-based ordering. For each ordering strategy, we evaluate two weighting schemes: (1) Factor-count-based weighting (our proposed approach), and (2) Original uniform weighting (as in AnoLLM).

The three causality-based orderings all consistently and significantly outperform the random ordering strategy (i.e.,

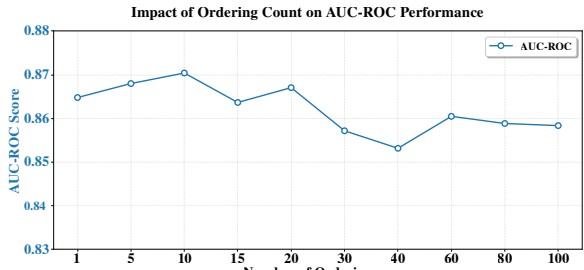

*Figure 4.* Effect of number of orderings on AUC-ROC performance on Fake job posts dataset.

without using any causal algorithm). Their results are nearly identical, which demonstrates that our method is not dependent on any specific causal discovery algorithm but is compatible with a variety of existing ones. Furthermore, the factor-count-based weighting strategy provides additional performance gains by appropriately emphasizing columns with stronger causal influence.

**Effect of Model Size.** Table 4 shows that performance is largely stable across model sizes on the Seismic dataset, with only minor differences among 135M, 360M, and 1.7B backbones. This indicates that our method is relatively stable under our current training setup and data scale. This may be because LLMs are primarily trained on text data, which has less relevance to tabular data. Therefore, it is recommended to use a relatively smaller LLM for tabular anomaly detection in our method.

**Effect of Number of Orderings.** As our column reordering module selects top-$K$ orders to better preserve causal relationships, we investigate the effect of the parameter $K$, which shows in Figure 4. As we can see, the model's performance does not change significantly as the number of orderings increases. As involving more orders will increase the testing time, it is suggested to use a moderate number of orders (around 10) in our method.

## 6. Conclusion

We propose CausalTAD, a method that injects causal knowledge into LLMs for tabular anomaly detection. It identifies causal relationships between columns and incorporates them through causal-driven ordering and reweighting strategies. Experiments on more than 30 datasets show that our method consistently outperforms state-of-the-art approaches across mixed-type and numerical settings. Our approach still relies on the capability of LLMs for factor extraction/annotation and introduces extra computational overhead from causal discovery and multiple orderings. Future work may focus on improving efficiency and robustness while retaining the causal benefits.

## Impact Statement

This work advances tabular anomaly detection by incorporating causal knowledge into LLM-based modeling. The method is intended for benign, high-stakes domains such as fraud detection and healthcare monitoring, where improved accuracy can support safer decision-making. As with any anomaly detection system, careful deployment and human oversight are necessary to mitigate potential misuse or over-reliance.

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

*Table 5.* Summary of datasets in our experiments (ODDS).

| Dataset | # points | # text | # num | # category | # Has column names | # outliers (%) |
|---|---|---|---|---|---|---|
| Annthyroid | 7,200 | 0 | 6 | 0 | No | 534 (7.42%) |
| Arrhythmia | 452 | 0 | 274 | 0 | No | 66 (15%) |
| BreastW | 683 | 0 | 9 | 0 | Yes | 239 (35%) |
| Cardio | 1,831 | 0 | 21 | 0 | Yes | 176 (9.6%) |
| Ecoli | 336 | 0 | 7 | 0 | Yes | 9 (2.6%) |
| ForestCover | 286,048 | 0 | 10 | 0 | Yes | 2,747 (0.9%) |
| Glass | 214 | 0 | 9 | 0 | No | 9 (4.2%) |
| Heart | 224 | 0 | 44 | 0 | No | 10 (4.4%) |
| Http (KDDCUP99) | 567,479 | 0 | 3 | 0 | No | 2,211 (0.4%) |
| Ionosphere | 351 | 0 | 33 | 0 | No | 126 (36%) |
| Letter Recognition | 1,600 | 0 | 32 | 0 | No | 100 (6.25%) |
| Lymphography | 148 | 0 | 3 | 15 | Yes | 6 (4.1%) |
| Mammography | 11,183 | 0 | 6 | 0 | No | 260 (2.32%) |
| Mulcross | 262,144 | 0 | 4 | 0 | No | 26,214 (10%) |
| Musk | 3,062 | 0 | 166 | 0 | No | 97 (3.2%) |
| Optdigits | 5,216 | 0 | 64 | 0 | No | 150 (3%) |
| Pendigits | 6,870 | 0 | 16 | 0 | No | 156 (2.27%) |
| Pima | 768 | 0 | 8 | 0 | No | 268 (35%) |
| Satellite | 6,435 | 0 | 36 | 0 | No | 2,036 (32%) |
| Satimage-2 | 5,803 | 0 | 36 | 0 | No | 71 (1.2%) |
| Seismic | 2,584 | 0 | 14 | 4 | Yes | 170 (6.5%) |
| Shuttle | 49,097 | 0 | 9 | 0 | No | 3,511 (7%) |
| Smtp (KDDCUP99) | 95,156 | 0 | 3 | 0 | No | 30 (0.03%) |
| Speech | 3,686 | 0 | 400 | 0 | No | 61 (1.65%) |
| Thyroid | 3,772 | 0 | 6 | 0 | No | 93 (2.5%) |
| Vertebral | 240 | 0 | 6 | 0 | Yes | 30 (12.5%) |
| Vowels | 1,456 | 0 | 12 | 0 | No | 50 (3.4%) |
| WBC | 278 | 0 | 30 | 0 | No | 21 (5.6%) |
| Wine | 129 | 0 | 13 | 0 | Yes | 10 (7.7%) |
| Yeast | 1,364 | 0 | 8 | 0 | Yes | 64 (4.7%) |

## A. Appendix Overview

This appendix provides supporting details for the empirical evaluation. It summarizes dataset characteristics used in our study (Sec. B) and reports the full AUC-ROC results over the ODDS benchmarks as well as the complementary F1 scores for the mixed-type benchmarks (Sec. C). It also documents the exact LLM prompts used for factor discovery and factor annotation, which underpin the causal discovery pipeline in Sec. 4.1.1.

## B. Dataset Statistics

Table 5 lists the datasets used in our experiments, covering ODDS benchmarks. It reports dataset size, the number of textual, numerical, and categorical columns, whether column names are available, and the proportion of anomalies. This table is intended to clarify the heterogeneity of the evaluation suite and the prevalence of text-bearing features.

**In handling missing column names and feature values**, we adopt and extend the conventions established by AnoLLM. For datasets without column names, we assign alphabetical placeholders (e.g., "A", "B", . . ., "AA", "AB"). To improve factor extraction in the causal discovery stage, we further increase the number of sampled instances presented to the LLM, which provides broader contextual coverage and helps the model identify dataset-specific factors more comprehensively. For missing feature values, we adhere to the same treatment as AnoLLM and map them to "Unknown". As shown in Table 5, multiple ODDS datasets contain missing names or values, and the detailed results in Table 6 show that our method still achieves SOTA performance under these conditions, further demonstrating the robustness of our approach.

## C. Additional Evaluation Results

Table 6 presents detailed AUC-ROC scores across the ODDS datasets and the mixed-type benchmarks. The table expands the main text by reporting method-by-dataset performance, including the AnoLLM-135M baseline and CausalTAD-135M

*Table 6.* Detailed AUC-ROC scores over 30 ODDS datasets and mixed-type benchmarks. AnoLLM uses the SmolLM-135M backbone; our method reports CausalTAD-135M.

| Datasets \ Methods | IForest | PCA | KNN | ECOD | DeepSVDD | RCA | SLAD | GOAD | NeuTral | ICL | DTE | REPEN | AnoLLM-135M | CausalTAD-135M |
|---|---|---|---|---|---|---|---|---|---|---|---|---|---|---|
| Annthyroid | 0.917 | 0.851 | 0.784 | 0.784 | 0.688 | 0.704 | 0.753 | 0.596 | 0.812 | 0.809 | 0.965 | 0.742 | 0.627 | 0.893 |
| Arrhythmia | 0.763 | 0.756 | 0.718 | 0.734 | 0.712 | 0.722 | 0.720 | 0.732 | 0.717 | 0.746 | 0.724 | 0.600 | 0.803 | 0.836 |
| BreastW | 0.500 | 0.753 | 0.753 | 0.737 | 0.401 | 0.753 | 0.640 | 0.402 | 0.273 | 0.616 | 0.791 | 0.639 | 0.997 | 0.997 |
| Cardio | 0.934 | 0.968 | 0.907 | 0.949 | 0.828 | 0.955 | 0.807 | 0.688 | 0.749 | 0.792 | 0.900 | 0.724 | 0.891 | 0.940 |
| Ecoli | 0.794 | 0.681 | 0.847 | 0.725 | 0.714 | 0.843 | 0.863 | 0.812 | 0.805 | 0.840 | 0.865 | 0.792 | 0.854 | 0.846 |
| ForestCover | 0.705 | 0.877 | 0.904 | 0.893 | 0.622 | 0.861 | 0.817 | 0.216 | 0.819 | 0.893 | 0.936 | 0.794 | 0.703 | 0.835 |
| Glass | 0.453 | 0.478 | 0.453 | 0.483 | 0.607 | 0.476 | 0.596 | 0.729 | 0.598 | 0.725 | 0.447 | 0.464 | 0.630 | 0.671 |
| Heart | 0.814 | 0.835 | 0.803 | 0.668 | 0.730 | 0.787 | 0.812 | 0.825 | 0.809 | 0.772 | 0.806 | 0.279 | 0.836 | 0.842 |
| Http (KDDCUP99) | 1 | 1 | 1 | 0.96 | 1 | 1 | 1 | 1 | 0.978 | 1 | 1 | 0.898 | 1 | 1 |
| Ionosphere | 0.841 | 0.881 | 0.933 | 0.761 | 0.964 | 0.910 | 0.954 | 0.926 | 0.954 | 0.966 | 0.980 | 0.650 | 0.127 | 0.252 |
| Letter Recognition | 0.316 | 0.302 | 0.400 | 0.454 | 0.417 | 0.445 | 0.632 | 0.307 | 0.344 | 0.355 | 0.380 | 0.363 | 0.510 | 0.510 |
| Lymphography | 0.673 | 0.826 | 0.860 | 0.830 | 0.899 | 0.919 | 0.964 | 0.817 | 0.847 | 0.827 | 0.909 | 0.808 | 0.968 | 0.993 |
| Mammography | 0.881 | 0.900 | 0.872 | 0.906 | 0.857 | 0.873 | 0.740 | 0.756 | 0.690 | 0.782 | 0.864 | 0.863 | 0.915 | 0.876 |
| Mulcross | 1 | 1 | 1 | 0.960 | 1 | 1 | 0.998 | 1 | 0.978 | 1 | 1 | 0.898 | 1 | 1 |
| Musk | 0.943 | 1 | 1 | 1 | 0.988 | 1 | 1 | 1 | 1 | 1 | 1 | 0.646 | 1 | 1 |
| Optdigits | 0.813 | 0.585 | 0.933 | 0.616 | 0.614 | 0.801 | 0.916 | 0.840 | 0.952 | 0.918 | 0.776 | 0.305 | 0.967 | 0.972 |
| Pendigits | 0.958 | 0.944 | 0.999 | 0.930 | 0.739 | 0.973 | 0.940 | 0.182 | 0.957 | 0.914 | 0.983 | 0.899 | 0.892 | 0.892 |
| Pima | 0.726 | 0.708 | 0.743 | 0.591 | 0.654 | 0.712 | 0.575 | 0.715 | 0.758 | 0.706 | 0.664 | 0.736 | 0.675 | 0.667 |
| Satellite | 0.777 | 0.666 | 0.817 | 0.622 | 0.718 | 0.730 | 0.801 | 0.752 | 0.786 | 0.794 | 0.765 | 0.690 | 0.789 | 0.843 |
| Satimage-2 | 0.991 | 0.982 | 0.997 | 0.971 | 0.957 | 0.998 | 0.999 | 0.993 | 0.960 | 0.995 | 0.993 | 0.997 | 1 | 1 |
| Seismic | 0.692 | 0.692 | 0.738 | 0.692 | 0.713 | 0.727 | 0.714 | 0.717 | 0.681 | 0.719 | 0.714 | 0.724 | 0.712 | 0.746 |
| Shuttle | 0.996 | 0.996 | 0.999 | 0.993 | 0.993 | 0.995 | 0.999 | 0.987 | 0.997 | 1 | 0.998 | 0.995 | 1 | 1 |
| Smtp (KDDCUP99) | 0.911 | 0.838 | 0.915 | 0.884 | 0.755 | 0.860 | 0.916 | 0.922 | 0.836 | 0.870 | 0.953 | 0.893 | 0.876 | 0.959 |
| Speech | 0.373 | 0.364 | 0.364 | 0.360 | 0.558 | 0.362 | 0.375 | 0.676 | 0.597 | 0.524 | 0.416 | 0.523 | 0.470 | 0.625 |
| Thyroid | 0.991 | 0.986 | 0.973 | 0.976 | 0.944 | 0.971 | 0.918 | 0.748 | 0.869 | 0.983 | 0.988 | 0.936 | 0.976 | 0.976 |
| Vertebral | 0.151 | 0.175 | 0.161 | 0.412 | 0.376 | 0.326 | 0.284 | 0.236 | 0.412 | 0.351 | 0.480 | 0.378 | 0.573 | 0.631 |
| Vowels | 0.595 | 0.528 | 0.826 | 0.615 | 0.562 | 0.785 | 0.798 | 0.728 | 0.645 | 0.765 | 0.836 | 0.617 | 0.670 | 0.731 |
| WBC | 0.998 | 0.994 | 0.989 | 0.998 | 0.930 | 0.983 | 0.996 | 0.298 | 0.650 | 0.957 | 0.739 | 0.983 | 0.966 | 0.969 |
| Wine | 0.385 | 0.438 | 0.707 | 0.743 | 0.683 | 0.747 | 0.902 | 0.125 | 0.218 | 0.277 | 0.508 | 0.808 | 0.758 | 1 |
| Yeast | 0.428 | 0.417 | 0.445 | 0.446 | 0.474 | 0.455 | 0.503 | 0.585 | 0.552 | 0.468 | 0.464 | 0.439 | 0.458 | 0.543 |
| Average | 0.744 | 0.747 | 0.794 | 0.756 | 0.736 | 0.789 | 0.798 | 0.677 | 0.741 | 0.779 | 0.795 | 0.681 | 0.788 | 0.835 |

under identical settings. Red denotes the best performance and blue denotes the second best. This comprehensive view highlights performance variability across datasets with different feature compositions.

Table 7 reports the F1 scores on the six mixed-type datasets for classical and deep-learning baselines, consistent with the evaluation protocol in the main paper. This provides a complementary perspective to AUC-ROC and supports the robustness of the comparative analysis on text-heavy benchmarks.

The prompt templates used for factor discovery and factor annotation are provided below. Consistent with the causal discovery procedure described in Sec. 4.1.1, the first prompt instructs the LLM to propose a diverse set of high-level, annotatable factors from serialized samples, while explicitly emphasizing text-heavy fields and enforcing a structured JSON schema. The second prompt operationalizes the annotation step by mapping each sample to discrete factor values under the previously defined criteria, ensuring consistent factor-value matrices for downstream causal discovery.

*Table 7.* F1 scores for all methods on the six mixed-type tabular datasets. **Bold** indicates the best performance. Results marked with † are cited from the original AnoLLM paper (Tsai et al., 2025).

| Methods \ Datasets | Fake job posts | Fraud ecommerce | Lympho-graphy | Seismic | Vehicle insurance | 20news groups | Average |
|---|---|---|---|---|---|---|---|
| *Classical methods* | | | | | | | |
| IForest† | 0.274 | 0.173 | 0.233 | 0.251 | 0.11 | 0.137 | 0.196 |
| PCA† | 0.256 | 0.209 | 0.567 | 0.266 | 0.124 | 0.133 | 0.259 |
| KNN† | 0.163 | **1** | 0.667 | 0.291 | 0.135 | 0.156 | 0.402 |
| ECOD† | 0.165 | 0.408 | 0.400 | 0.282 | 0.112 | 0.132 | 0.250 |
| *Deep learning based methods* | | | | | | | |
| DeepSVDD† | 0.136 | **1** | 0.567 | 0.258 | 0.115 | 0.152 | 0.371 |
| RCA† | 0.137 | **1** | 0.667 | 0.32 | 0.135 | 0.129 | 0.398 |
| SLAD† | 0.175 | 0.988 | 0.667 | 0.285 | 0.155 | 0.159 | 0.405 |
| GOAD† | 0.129 | 0.92 | 0.667 | 0.295 | 0.119 | 0.136 | 0.378 |
| NeuTral† | 0.115 | **1** | 0.633 | 0.195 | 0.12 | 0.195 | 0.376 |
| ICL† | 0.245 | **1** | 0.667 | 0.298 | 0.108 | 0.19 | 0.418 |
| DTE† | 0.107 | **1** | 0.667 | 0.239 | 0.121 | 0.185 | 0.387 |
| REPEN† | 0.164 | **1** | 0.667 | 0.306 | 0.126 | 0.124 | 0.398 |
| *AnoLLM* | | | | | | | |
| SmolLM-135M† | 0.325 | **1** | 0.767 | 0.279 | 0.091 | **0.241** | 0.462 |
| SmolLM-360M† | 0.343 | 0.992 | 0.8 | 0.336 | 0.108 | 0.22 | 0.478 |
| *CausalTAD (Ours)* | | | | | | | |
| SmolLM-135M | 0.402 | **1** | 0.833 | **0.465** | 0.162 | **0.241** | 0.505 |
| SmolLM-360M | **0.405** | 0.992 | **1** | 0.447 | **0.174** | 0.22 | **0.529** |

# D. LLM Prompts

### Job Posting Factor Discovery

**Task:** Job Posting Factor Discovery
**Task Background**
This is a job posting dataset (online recruitment ads).
A single job advertisement constitutes each sample, featuring structured attributes such as location, employment type, required experience, required education, industry, and function, alongside multiple rich-text sections including title, company profile, description, requirements, and benefits.
**Important Notes:**

- All samples shown below should be treated as NORMAL job posts for the purpose of factor discovery.

- Do NOT try to detect fraud / anomalies.

- Your goal is to describe the job posting in a comprehensive, structured way based on observable information.

**Text-heavy dataset reminder (very important):** Many key signals are in the text columns. You must extract semantic factors mainly from:

- title

- company_profile

- description

- requirements

- benefits

**Recruitment-post perspective:** A job posting typically describes: employer/brand, role scope, responsibilities, required skills & experience, education, employment type, compensation signals, benefits, work mode (telecommuting/remote), and other application or screening signals. Your factors should help reconstruct the full job-post information landscape.

**Dataset Description (Column Distribution / Statistics)**

1. job_id: Unique identifier for each job posting

2. title: Text column, not analyzed statistically.

3. location: Text column, location values have small and similar proportions to each other.

4. department: Text column, 64.85% missing values; among valid data, Sales, Engineering, Marketing, Operations, IT account for 35.1% combined.

5. salary_range: Text column, highly diverse.

6. company_profile: Text column.

7. description: Text column.

8. requirements: Text column.

9. benefits: Text column.

10. telecommuting: Categorical column (2 classes): 0=95.7%, 1=4.2%. Extremely imbalanced, mainly 0.

11. has_company_logo: Categorical column (2 classes): 0=18.1%, 1=81.8%. Imbalanced, mainly 1.

12. has_questions: Categorical column (2 classes): 0=49.9%, 1=50%. Very balanced.

13. employment_type: Categorical column: blank 18.9%, Contract 8.8%, Full-time 65.4%, Other 1.2%, Part-time 4.3%, Temporary 1.1%. Dominated by Full-time, followed by blank, other types have small proportions (especially Other/Temporary).

14. required_experience: Categorical column (8 classes): blank 39%, Associate 13.4%, Director 2.1%, Entry level 14.9%, Executive 0.7%, Internship 2.3%, Mid-Senior level 21.3%, Not Applicable 5.9%.

15. required_education: Categorical column (14 classes): blank 45%; Bachelor's 29.5%; High School 11%; Associate 1.6%; Unspecified 8%; Certification 0.9%; Professional 0.3%; Some College 0.6%; Doctorate 0.1%; vocational education (Vocational, etc.) totals about 2.44%. Overall: blank and Bachelor's/High School account for the majority; among non-blank data, higher education (Bachelor's/Master's/Doctorate/Professional) accounts for 58.69%, vocational education is minimal ($\approx 2.44\%$).

16. industry: Categorical column: blank $\approx 10\%$, Information Technology and Services $\approx 10\%$, Computer Software $\approx 8\%$, others are smaller and relatively evenly distributed.

17. function: Categorical column, blank 35.71%; among non-blank: Information Technology 10.17%, Sales 8.4%, Engineering 7.46%, Customer Service 7.05%; grouped by function, among non-blank: technical roles 30.17%, business roles 23.99%, operations roles 18.34%.

**Current Task**

You will see multiple "job posting" samples. Based on these samples, extract semantic factors that can comprehensively describe a job posting.

Here, "job posting descriptive factors" refer to: abstract factors across dimensions such as employer/brand, role scope, responsibilities, required skills & experience, education, employment type, compensation signals, benefits, work mode (telecommuting/remote), and application/screening signals. You must NOT and are NOT allowed to target "fraud/anomaly detection"; your goal is to "clearly describe, decompose, and structure" the job posting information.

**Text Column Priority (Very Important)**

This dataset has many text columns, with key information primarily in:

- title (job title)

- company_profile (company introduction)

- description (job description/responsibilities)

- requirements (job requirements)

- benefits (compensation and benefits)

You MUST extensively extract semantic information from these text columns (e.g., keywords/phrases/sentence patterns implying scope of duties, skill requirements, benefit structure, work intensity, application methods, etc.). Do not rely solely on structured columns.

**Factor Definition Requirements**

1. Semantic abstraction: Factors should be interpretable abstract concepts, not direct copies of raw column names. Joint factors are allowed and encouraged: determined by multiple columns (especially combinations of text + categorical columns). If it's a joint factor, clearly specify the value determination rules.

2. Multi-valued: Factor values must be multiple discrete numerics (e.g., 0,1,2,-1). Try to make the values rich.

3. Annotability: Factors must be clearly annotatable from sample data (especially from text columns via rules).

4. Information coverage: Factors should cover different dimensions of job posting information (company, role, responsibilities, requirements, benefits, work mode, application/screening signals, etc.).

5. Detailed criteria: annotation_criteria must be extremely detailed, written as step-by-step judgment rules that can guide a weaker model to annotate samples item by item (e.g., if-then/priority lists). Note: Rules should stabilize annotation, but do NOT require outputting reasoning processes; annotation phase only outputs result JSON.

6. Diversity (dimensional coverage): Factor quantity should be as rich as possible. Don't worry about redundancy; subsequent steps will filter.

**Missing/Unknown Handling Convention (Must Follow)**

In this dataset, structured columns may have missing values, potentially explicitly written as the string 'unknown' (or blank). Additionally, text columns may not mention certain types of information at all.

To ensure consistency in subsequent automatic annotation:

1. Each factor must reserve 0 as a special value:

   - 0 = not mentioned / insufficient information / field is unknown or blank (cannot reliably determine from title/company_profile/description/requirements/benefits or structured columns).

2. Other values (1,2,3...) represent "clearly determinable" categories/levels.

3. annotation_criteria must explicitly include rules for "how to determine as 0" (e.g., related field is unknown/blank, and text also shows no corresponding signal → 0).

**Please explicitly cover the following dimensions (try to produce multiple factors for each dimension):**

- Basic role dimensions: job category/functional direction/department affiliation/industry direction/job title type

- Level and experience dimensions: semantic mapping and consistency of seniority, required_experience, required_education

- Responsibility content dimensions (mainly text extraction): core responsibility types (operations/sales/R&D/customer service/management, etc.), work objects/scenarios, deliverables

- Skill requirement dimensions (mainly text extraction): tech stack/tools/soft skills/certifications/language requirements

- Compensation and benefit signals (text+salary_range): whether salary is explicit, salary range format, whether there are performance/bonus signals

- Benefit structure dimensions (mainly text extraction): benefit richness, benefit types (insurance/leave/flexibility/training/stock, etc.)

- Work mode dimensions: telecommuting/remote tendency, full-time/part-time/contract, etc.

- Recruitment and screening signals: has_questions, whether application materials are emphasized, whether interview/assessment/video are mentioned

- Text quality/completeness dimensions: description length level, requirement itemization degree, whether benefit description is missing, etc.

**Focus Factor Selection Criteria (Important)**

Among the extracted factors, recommend 2-5 as focus_factor (core target factors), with the following criteria:

1. Value diversity: Expected to have at least 3 different values in the current data

2. Semantic relevance: The factor has strong potential structural associations with other factors (e.g., influencing how other parts of the job posting are written/content)

3. Examples:

   - "Job responsibility type" (0=technical R&D/1=sales business/2=operations marketing/3=customer support/4=management/5=other)
   - "Benefit richness level" (0=none/1=few/2=medium/3=many)
   - "Text completeness level" (0=very short and missing key info/1=average/2=complete and detailed/3=very detailed and structured)
   - Factors that are almost constant or have only a single value

**Joint Factor Examples**

**Example 1: Remote-Job Type Alignment (joint)**

- Based on columns: [telecommuting, title/description]

- Possible values: [0=consistent and clear, 1=weakly consistent, 2=contradictory/unclear]

- Rules: If telecommuting=1 and text explicitly states remote/anywhere=0; telecommuting=1 but text contains many on-site/travel/in-office requirements=2; otherwise=1.

**Example 2: Requirement Itemization Level (text)**

- Based on columns: [requirements]

- Possible values: [0=none or very little, 1=prose style, 2=itemized, 3=itemized with clear grouping]

- Rules: Whether bullets/numbering/section headers (Must/Preferred/Bonus) appear.

**Output Format (Strictly Follow JSON Format)**

```
{
  "factors ": {
    "factor_name_1": {
      " description ": "Detailed  description  of  the  factor ",
      " possible_values ": [0,  1,  2],
      " annotation_criteria ": "How to determine  the  factor  value  from samples ( detailed ,  executable  step−by−step
          judgment rules ; do not  require  outputting  reasoning  process )",
      "column_based": ["ColumnName1", "ColumnName2"]
    }
  },
```

```
    "recommended_focus_factors": [
      {
        "factor_name": "factor_name_1",
        "reasoning": "Reason for recommendation",
        " expected_value_diversity ": "Expected to have 3–4 different values in the data"
      }
    ]
}
```

**Special Notes**

- Factor names use English, descriptions use English

- possible_values can only be numbers

- annotation_criteria must be sufficiently detailed (usable for subsequent automatic annotation)

- column_based explicitly lists the column names your factor is primarily based on; explicitly write out text columns

- Do not guide "fraud/anomaly" judgment

**Sample Data**
Sample 1
title is Marketing Manager
location is AU, Victoria, Melbourne
department is Marketing
salary_range is unknown
company_profile is We are a leading company in consumer goods industry seeking a talented marketing professional to join our dynamic team.
description is Develop and execute comprehensive marketing strategies to drive brand awareness and customer engagement. Lead cross-functional teams to launch new products and campaigns.
requirements is Bachelor's degree in Marketing or related field. 5+ years of experience in marketing management. Strong analytical and communication skills.
benefits is Competitive salary package, health insurance, flexible working arrangements, professional development opportunities.
telecommuting is 0
has_company_logo is 1
has_questions is 1
employment_type is Full-time
required_experience is Mid-Senior level
required_education is Bachelor's Degree
industry is Consumer Goods
function is Marketing
Sample 2
...
Please carefully analyze the above samples and extract multiple high-quality semantic factors, recommending suitable focus_factor(s).

**Full Annotation - Complete English Version**

**Prompt 2: Full Annotation - Complete English Version**
**Task:** Job Posting Sample Factor Annotation
**Task Background**
This is a job posting dataset (online recruitment ads).
Each sample is a single job advertisement with structured fields (e.g., location, employment_type, re-

quired_experience, required_education, industry, function) and multiple rich text fields (e.g., title, company_profile, description, requirements, benefits).

**Important Notes:**

- All samples shown below should be treated as NORMAL job posts for the purpose of factor discovery.

- Do NOT try to detect fraud / anomalies.

- Your goal is to describe the job posting in a comprehensive, structured way based on observable information.

**Text-heavy dataset reminder (very important):** Many key signals are in the text columns. You must extract semantic factors mainly from:

- title

- company_profile

- description

- requirements

- benefits

**Recruitment-post perspective:** A job posting typically describes: employer/brand, role scope, responsibilities, required skills & experience, education, employment type, compensation signals, benefits, work mode (telecommuting/remote), and other application or screening signals. Your factors should help reconstruct the full job-post information landscape.

**Factor Definitions**

The following factors and their annotation criteria have been defined:

factors_json

**Sample to Annotate**

sample_text

**Annotation Requirements**

Please strictly annotate this sample according to the annotation_criteria of the above factors.

**Text Column Reminder (Very Important)**

The sample contains multiple long text fields (title/company_profile/description/requirements/benefits). When annotating, you MUST read these texts and determine factor values based on the rules.

**Output Format (Must be "Pure JSON Text")**

You must output a JSON object (starting with { and ending with }). Do not output any explanatory text; do not output Markdown; do not use code blocks (no "' should appear).

Example (structure only):

```
{
  "factor_name_1": 0,
  "factor_name_2": 2
}
```

**Important Notes**

1. Each factor's value must be within its possible_values range

2. Unknown/missing/not mentioned - unified determination:

   - Data may contain the string 'unknown' (or blank) to indicate missing; text columns may also completely fail to mention relevant information.

   - When you cannot reliably determine from any relevant column (especially title/company_profile/description/requirements/benefits):

     – Default output 0 (indicating "not mentioned/insufficient information/unknown").

– Only when the factor's possible_values explicitly includes -1, you can use -1 to indicate "unknown"; otherwise always use 0.

3. Do not output text like "unknown"; all values must be numbers (integers)

4. Do not output reasoning process; only output the JSON object itself, without additional explanatory text

Please begin annotation:

