# OpenReview forum: "CausalTAD: Injecting Causal Knowledge into Large Language Models for Tabular Anomaly Detection"
_ICML.cc/2026/Conference — Submitted to ICML 2026_

### Official Review · Reviewer_ik6q · 2026-02-24

**Soundness:** 2
**Presentation:** 2
**Significance:** 3
**Originality:** 2
**Overall Recommendation:** 4
**Confidence:** 4

**Summary:**

This paper focuses on the study of tabular anomaly detection via LLMs. This paper follows the model structure of existing method AnoLLM, and introduces two modifications: causal relationships between columns and reweighting different columns.

**Compliance With Llm Reviewing Policy:**

Affirmed.

**Final Justification:**

Though the main concerns have been solved, the authors should incorporate the following parts into the final manuscript: (i) the four related baselines as discussed in W5, and the corresponding experimental results, (ii) the theoretical analysis. And I'm thus raising my score to weak accept accordingly.

**Key Questions For Authors:**

Please see the weaknesses above.
Main concerns have been solved.

**Limitations:**

No limitations discussed. The authors could include Fundamental contradiction, reliance on black-box LLMs for causal discovery, privacy and data sensitivity, reliance on semantic information,..., as discussed in weaknesses above.

**Strengths And Weaknesses:**

Strengths:

S1: The problem is important.
S2: Introducing LLMs for tabular anomaly detection is appealing.

Weaknesses:

W1: Fundamental contradiction.
The paper's central premise is that imposing a fixed, causal-driven column order during serialization helps the LLM model inter-column dependencies. This directly conflicts with the well-established property that tabular data is permutation-invariant with respect to columns [1]. A robust model for tabular data should be invariant to column order.

[1] Making Pre-trained Language Models Great on Tabular Prediction. ICLR 2024

W2: The order-aware learning is brittle.
By training on samples serialized with a specific set of orders (even multiple ones), the model is implicitly learning to exploit this fixed structure. If the causal graph inferred from normal data is brittle or only partially correct, forcing all test samples into that order could actively harm detection performance on anomalies that violate this assumed structure.

W3: Internal contradiction. Is column order important or not? The paper's logic is internally inconsistent about the role of column order.

* Step 1 (Causal Discovery, Sec 4.1.1): The authors serialize each sample as "c1 is x1, c2 is x2, ... , cd is xd" using the original column order from the dataset (which could be alphabetical, random, or whatever). They then feed this into COAT and assume the LLM can correctly extract causal factors from this arbitrarily ordered text. The implicit assumption here is that LLMs are robust to column order—they can understand the data and its relationships regardless of which column comes first.

* Step 2 (Training, Sec 4.1.3): After discovering the "causal" orderings, the authors claim these orders are important. They force all training samples to be serialized using these specific orders, and train the LLM to predict columns sequentially. The implicit assumption here is that column order matters—the LLM should learn to exploit this structure.

The paper provides no explanation for how the LLM can magically be both order-sensitive and order-insensitive as needed. This methodological self-contradiction calls the entire framework into question.

W4: Unsoundness of the causal discovery pipeline:

* Lack of theoretical grounding for factor extraction: The entire causal graph hinges on the factors extracted by an LLM. The paper offers no theoretical guarantee that these factors are meaningful, stable, or correctly represent the underlying data-generating process. The process is essentially a black box. How do you know the extracted factors are not artifacts of the prompt or the LLM's biases?

* Can not generalize to settings without semantic meaning.  For purely numerical datasets with generic column names (e.g., "A", "B", "C") or no column names, what semantic factors can an LLM extract? This calls into question the applicability of the method to a wide range of datasets where column names lack rich semantic meaning.

* Privacy concerns: The causal discovery phase requires sending the whole serialized samples to an external, closed-source LLM (GPT-4). This is a critical limitation for many real-world applications involving sensitive data, severely restricting the method's practical deployment.

W5: Insufficient baselines. Lack of recent tabular anomaly detection baselines. For example, modern nerual network baselines: MCM[2], DRL[3], NPT[4]. Recent LLM-based method: LLM-DAS[5]. If additional comparison is time consuming, at least, they should be described in related work.

[2] MCM: Masked Cell Modeling for Anomaly Detection in Tabular Data. ICLR 2024

[3] DRL: Decomposed Representation Learning for Tabular Anomaly Detection. ICLR 2025

[4] Beyond Individual Input for Deep Anomaly Detection on Tabular Data. ICML 2024

[5] LLM as an Algorithmist: Enhancing Anomaly Detectors via Programmatic Synthesis. ICLR 2026

W6: Lack clarity of causal discovery algorithms. It is hard to understand without equations to identify the input and output of causal discovery algorithms. And there are no discussion about what these algorithms do.

W7: Experiments are insufficient.
* What is the performance of directly querying LLM to complete column ordering, rather than first extract factors and then reorder the columns?
* What is the results of using different types of LLMs?
* Is there any failure case?

---

> ### Author Rebuttal · Authors · 2026-03-31
>
> # response to ik6q
>
>
> # W1
> The permutation-invariance argument applies to discriminative models that observe all columns at once, whereas our method follows an autoregressive anomaly-detection paradigm. In our setting, the model scores each column via $p(x_i \mid x_{<i})$, so column order directly affects the available context. If the order is random, key parent features may appear too late, leading to missing information, noisy likelihood estimates, and more false positives. Therefore, a reasonable column ordering is essential for effective anomaly detection.
>
> # W2
> The causal discovery framework COAT provides strong theoretical guarantees. We will provide a detailed explanation in W6.
>
> # W3
>
> As noted earlier, column ordering matters only in the autoregressive (AR) anomaly detection paradigm: COAT is a mature framework that uses row-wise context at input time, so column order is not important in Step1, whereas Step2 performs autoregressive likelihood factorization and remains order-dependent because prefix context matters.
>
> # Added Baselines for W5
>
> ## comparison on datasets with free-text columns
> Comparisons on the **fakejob** dataset. The key metric, **AUC-ROC**, is as follows:
>
> | Method | AUC-ROC |
> |---|---:|
> | MCM | 0.676 |
> | DRL | 0.611 |
> | NPT-AD | 0.593 |
> | LLM-DAS (DRL) | 0.637 |
> | AnoLLM | 0.814 |
> | LLM-DAS (AnoLLM) | 0.846 |
> | CausalTAD-135M | **0.873** |
>
> **Conclusion:**
> - In the "**mixed-type datasets with free-text columns**" scenario we focus on, CausalTAD significantly outperforms the other recent baselines
> - We tried using the LLM-DAS data augmentation method to enhance our AnoLLM. It still remains weaker than our CausalTAD
>
> ## Comparison on datasets purely numerical and categorical datasets
> Results are taken directly from the original paper on the same datasets as ours(including BreastW, Wine, Arrhythmia, Mammography, Optdigits, Pendigits, Satimage-2, Shuttle, Thyroid, WBC, Satellite).
>
> | Dataset | DRL | MCM | LLM-DAS (PCA) | LLM-DAS (DRL) | NPT-AD | CausalTAD-135M |
> |---|---:|---:|---:|---:|---:|---:|
> | average | 0.957 | 0.946 | 0.732 | 0.958 | 0.941 | 0.942 |
>
> Although CausalTAD is indeed weaker than some recent baselines on certain datasets, the average gap is less than 0.02.
>
>
>
> # W6
>
> The causal discovery framework COAT used in our paper follows an iterative **propose-parse-discover-feedback** loop. Given a small subset $\tilde{\mathcal{D}} \subset \mathcal{D}$, the LLM $\Psi$ proposes candidate factor mappings $\mathcal{W}^t=\Psi(p^t,\tilde{\mathcal{D}})$, and the parser $\Psi_a$ extracts structured factors $z_i=w_i(x):=\Psi_a(x,w_i,p_p)$, forming $h_{\leq t}(X)$. Then the causal discovery algorithm $\mathcal{A}$ (e.g., FCI) infers the current graph as $\mathcal{G}^t=\mathcal{A}(\mathbf{Z}^{\leq t}\cup\{Y\})$. For feedback, the next-round subset is selected by maximizing conditional entropy,
> $$
> \tilde{\mathcal{D}}^{t+1}=\arg\max_{\tilde{\mathcal{D}}\subset \mathcal{D}} H_{\tilde{\mathcal{D}}}\!\left(Y \mid h_{\leq t}(X)\right),
> $$
> so the LLM is guided to focus on residual information rather than directly guessing a black-box causal graph.
>
>
> # W4
>
> ## Theoretical guarantees
>
> If the LLM finds a new valid factor $w_{k+1}$ in the feedback stage (i.e., $Y \not\perp w_{k+1}(X) \mid h_{[k]}(X)$), then $I(Y; X \mid h_{[k+1]}(X)) < I(Y; X \mid h_{[k]}(X))$; moreover, under data faithfulness, the unexplained proportion satisfies $$ \frac{I(Y; X \mid h_{\leq t}(X))}{I(Y; X)} \leq \left(\frac{1}{1-C_{\Psi}}\right)^{-tp - z_{\delta}\sqrt{tp(1-p)}} $$, which implies that the extracted factor set converges to an effective Markov Blanket of $Y$ rather than arbitrary heuristic features.
>
>
> ## generalize to settings without semantic meaning
>
> The original COAT code shows that factor extraction does not rely only on column names; instead, it samples a small subset so the language model can understand the data content and type, and the **Feedback Construction** step further improves extraction for unnamed columns by selecting the subset with the highest conditional entropy to enrich the next-round prompt.
>
> ## Privacy concerns
>
> The factor proposal stage sends only a very small number of maximum-entropy samples to a closed-source LLM such as GPT-4, while the factor parsing stage can be done by a locally deployed open-source small model, so only a few samples and column names are exposed.
>
>
>
> # W7: Experiments are insufficient
>
>
> To evaluate directly relying on the LLM for column ordering and reweighting, we ran single-round prompting experiments: although random order reached 0.800 AUC-ROC, LLM-generated orders varied across models (GPT-4: 0.723, GPT-5.4: 0.820, Gemini 3.1 Pro: 0.798), and even with extra manually summarized observations the gains were limited, while CausalTAD (GPT-4) achieved 0.873, showing that explicit causal ordering is more stable and effective than prompt-based ordering.

---

> > ### Author Rebuttal · Reviewer_ik6q · 2026-04-02
> >
> > Though the main concerns have been solved, the authors should incorporate the following parts into the final manuscript: (i) the four related baselines as discussed in W5, and the corresponding experimental results, (ii) the theoretical analysis. And I'm thus raising my score accordingly.

---

> > > ### Author Response · Authors · 2026-04-06
> > >
> > > Thank you very much for acknowledging that the main concerns have been adequately addressed and for your constructive suggestions. We also appreciate your positive reassessment of our work.
> > >
> > > Regarding your helpful request, we would like to respectfully note that the official notification states that authors are not allowed to upload a revised manuscript during the rebuttal period, and can only submit an updated version in the final camera-ready stage if the paper is accepted. We have taken careful note of your suggestions and will incorporate the four additional baselines in W5, their corresponding results, as well as the theoretical analysis, into the final version.

---

### Official Review · Reviewer_9yMb · 2026-03-12

**Soundness:** 3
**Presentation:** 2
**Significance:** 2
**Originality:** 2
**Overall Recommendation:** 3
**Confidence:** 1

**Summary:**

CausalTAD proposes injecting causal knowledge into LLM-based tabular anomaly detection by reordering columns according to causal relationships prior to serialization. The method first uses an LLM (via the COAT framework) to extract high-level latent factors from tabular data, constructs a factor-level causal graph using off-the-shelf discovery algorithms, and then projects this graph onto a column-level preference matrix. Column ordering is formulated as a Linear Ordering Problem (LOP), and a complementary reweighting scheme emphasizes columns that map to more causal factors. Experiments across 36 datasets show consistent improvements over the primary baseline, AnoLLM.

**Compliance With Llm Reviewing Policy:**

Affirmed.

**Key Questions For Authors:**

1-	Could the authors justify why directly prompting an LLM to suggest a causal column ordering and reweighting was not considered as a baseline, given the significant complexity of the proposed pipeline?
2-	Does the quality of causal factor extraction by the LLM degrade when there are many columns?

**Limitations:**

The causal interpretation is not fully justified, as the factors are generated by LLMs and the projection from factor-level relations to column-level dependencies is heuristic. In addition, the method introduces notable computational overhead due to reliance on external LLMs, and the empirical evaluation lacks rigor in several aspects, including the absence of statistical significance testing and limited comparison with recent baselines.

**Strengths And Weaknesses:**

***Strength***
1- The ablation in Table 3 demonstrates that results are stable across PC, LiNGAM, and FCI, suggesting the method is not brittle to the choice of causal discovery algorithm. This is a meaningful robustness result.


***Weakness***
1- The method is untested on high-dimensional tabular data with many columns, and scalability to such settings remains an open question.
2- The causal-aware reweighting uses simple factor counts as a proxy for causal importance (Equation 4). This is a coarse heuristic.
3- Minor Presentation Issue: The hyperlink for the Figure 3 in line 365 incorrectly redirects to Table 2 rather than Figure 3.

---

> ### Author Rebuttal · Authors · 2026-03-31
>
> # Response to Reviewer 9yMb
>
>
> # Regarding Weaknesses
>
> ## W1
>
> We added an experiment with the ODDS Speech dataset, which have **400 columns**. The results are shown below. **Our method still achieve SOTA performance**.
>
> | Dataset | AnoLLM-135M | CausalTAD-135M |
> | --- | ---: | ---: |
> | Speech | 0.470 | 0.625 |
>
>
> ## W2
> Although this method is indeed coarse, it is still practically feasible; our experiments on multiple datasets have demonstrated this.
>
>
> # Regarding Limitations
>
> ## The causal interpretation is not fully justified
>
> While factors are generated with LLM assistance, our causal claim is not based on unconstrained graph guessing. COAT follows an iterative **propose–parse–discover–feedback** loop: the LLM proposes candidate factors, a parser converts them into structured variables, and a causal discovery method (e.g., FCI) updates the graph; entropy-based feedback then focuses the next round on residual information. Theoretical results (Prop. 2.4/2.5) show that when a newly extracted factor remains conditionally informative, unexplained mutual information strictly decreases, and the residual ratio $I(Y;X\mid h_{\le t}(X))/I(Y;X)$ decays exponentially with high probability. This indicates convergence toward an effective Markov blanket of $Y$, rather than arbitrary heuristic features. Full derivations are provided in *Discovery of the Hidden World with Large Language Models*.
>
>
> ## the method introduces notable computational overhead due to reliance on external LLMs
>
> We acknowledge the concern, but most COAT computation is dataset-level offline preprocessing rather than training-time or online inference work. Factor extraction and column ordering are one-time steps per dataset and do not need to be repeated. This stage uses no GPU, so for multiple datasets it can run in parallel with LLM fine-tuning to reduce total time. Since COAT does not alter the model architecture, it adds no inference-time overhead in deployment.
>
>
> ## the empirical evaluation lacks rigor in several aspects
>
> Thank you for the reminder. Regarding the comparison with the latest baselines, we have provided a detailed comparison in the "Added Baselines" section. As for the empirical evaluation, because rebuttal time is limited, we will add this important result in the revised paper later.
>
>
> # Regarding Key Questions
>
> ## Q1
>
> To evaluate directly relying on the LLM for column ordering and reweighting, we ran single-round prompting experiments: although random order reached 0.800 AUC-ROC, LLM-generated orders varied across models (GPT-4: 0.723, GPT-5.4: 0.820, Gemini 3.1 Pro: 0.798), and even with extra manually summarized observations the gains were limited, while CausalTAD (GPT-4) achieved 0.873, showing that explicit causal ordering is more stable and effective than prompt-based ordering.
>
>
> ## Q2
>
> Our causal discovery framework provides an **exponential convergence guarantee (Proposition 2.4 & 2.5)**.
> After $t$ rounds of iteration, the proportion of information that cannot be explained by the extracted factors decays **exponentially** with probability at least $1-\delta$:
> $$ \frac{I(Y; X | h_{\leq t}(X))}{I(Y; X)} \leq \left(\frac{1}{1-C_{\Psi}}\right)^{-tp - z_{\delta}\sqrt{tp(1-p)}} $$
> Therefore, even for high-dimensional data, the framework can still complete factor extraction with high quality in only a few iterations.
>
>
>
> # Added Baselines
>
> ## comparison on datasets with free-text columns
> Comparisons on the **fakejob** dataset. The key metric, **AUC-ROC**, is as follows:
>
> | Method | AUC-ROC |
> |---|---:|
> | MCM | 0.676 |
> | DRL | 0.611 |
> | NPT-AD | 0.593 |
> | LLM-DAS (DRL) | 0.637 |
> | AnoLLM | 0.814 |
> | LLM-DAS (AnoLLM) | 0.846 |
> | CausalTAD-135M | **0.873** |
>
> **Conclusion:**
> - In the "**mixed-type datasets with free-text columns**" scenario we focus on, CausalTAD significantly outperforms the other recent baselines
> - We tried using the LLM-DAS data augmentation method to enhance our AnoLLM. It still remains weaker than our CausalTAD
>
> ## Comparison on datasets purely numerical and categorical datasets
> Results are taken directly from the original paper on the same datasets as ours(including BreastW, Wine, Arrhythmia, Mammography, Optdigits, Pendigits, Satimage-2, Shuttle, Thyroid, WBC, Satellite).
>
> | Dataset | DRL | MCM | LLM-DAS (PCA) | LLM-DAS (DRL) | NPT-AD | CausalTAD-135M |
> |---|---:|---:|---:|---:|---:|---:|
> | average | 0.957 | 0.946 | 0.732 | 0.958 | 0.941 | 0.942 |
>
> Although CausalTAD is indeed weaker than some recent baselines on certain datasets, the average gap is less than 0.02.

---

> > ### Author Rebuttal · Reviewer_9yMb · 2026-04-02
> >
> > Thank you for the detailed rebuttal. I appreciate the additional experiments and clarifications.
> >
> > The comparison with direct LLM-based ordering (Q1) is particularly helpful and strengthens the motivation for the proposed pipeline. The discussion on the COAT framework and its theoretical properties also improves the justification of the causal interpretation.
> >
> > However, some concerns remain only partially addressed. The scalability claim is supported by a single additional dataset without broader evaluation or runtime analysis. The causal-aware reweighting remains heuristic without deeper justification or ablation. Additionally, empirical rigor is still limited, particularly the lack of statistical significance testing and reliance on results taken from prior work for some baselines.
> >
> > Overall, while the rebuttal improves clarity and addresses some questions, I believe the paper would benefit from further empirical validation and stronger justification of key design choices.
> >
> > I keep my original score.

---

> > > ### Author Response · Authors · 2026-04-06
> > >
> > > Thank you for the follow-up questions.
> > >
> > > ## For High-Dimensional Tabular Datasets
> > >
> > > ### Time Cost
> > >
> > > The time cost is concentrated in two parts: 1. causal discovery and 2. column ordering.
> > >
> > > **1. Causal Discovery**
> > >
> > > The causal discovery stage requires one model call per sample, with the total input and output length under 4k. For GPT-4o, this takes 2 to 4 seconds (source: https://artificialanalysis.ai/models/gpt-4o). We conservatively use 5 seconds for estimation and account for API concurrency.
> > >
> > > If the dataset contains $n$ samples and the parallel concurrency level is $p$, then the theoretical time is:
> > >
> > > $$
> > > T(n,p)=\left\lceil \frac{n}{p}\right\rceil \cdot t_s,\qquad t_s=5\text{ s}
> > > $$
> > >
> > > For the highest-dimensional tabular dataset in the ODDS library, Speech: $n=3686$, $m=400$, the theoretical runtime under different concurrency levels is:
> > >
> > > | Concurrency $p$ | Number of batches $\lceil 3686/p\rceil$ | Theoretical runtime |
> > > |---|---:|---:|
> > > | 1 | 3686 |5 hours 7 minutes |
> > > | 4 | 922 | 1 hour 16 minutes |
> > > | 8 | 461 | 38 minutes |
> > > | 16 | 231 | 19 minutes |
> > > | 32 | 116 | 9 minutes |
> > >
> > > **2. column ordering**
> > >
> > > The core issue is the Linear Ordering Problem (LOP). Since it is NP-hard, a polynomial-time exact worst-case upper bound is not expected.
> > >
> > > Our method does not aim to find the absolute optimal ordering in every case, but rather a sufficiently good ordering within a fixed time budget. On the 400-column Speech dataset, we evaluated different search budgets:
> > >
> > > | Time Limit | AUC-ROC |
> > > |---|---:|
> > > | AnoLLM baseline | 0.470 |
> > > | 60 s | 0.593 |
> > > | 300 s | 0.601 |
> > > | 600 s | 0.621 |
> > > | 1200 s | 0.623 |
> > > | 2400 s | 0.622 |
> > > | 3600 s | 0.624 |
> > > | Full search | 0.625 |
> > >
> > > A budget of around 600 seconds is already enough to reach performance close to full search.
> > >
> > > **3. Conclusion**
> > >
> > > Even for a 400-dimensional dataset, the method can finish in about **50 minutes** total. In an enterprise setting, for example with DeepSeek-R1-Distill-Qwen-32B and higher concurrency, the total time cost can be reduced to **within 20 minutes**.
> > >
> > >
> > > ### Performance
> > >
> > > We further validated the method on more high-dimensional datasets. The AUC-ROC results are:
> > >
> > > | Dataset | Number of Columns | CausalTAD | AnoLLM |
> > > | --- | ---: | ---: | ---: |
> > > | Speech | 400 | 0.625 | 0.470 |
> > > | Musk | 166 | 1.000 | 1.000 |
> > > | Arrhythmia | 274 | 0.836 | 0.803 |
> > >
> > > On high-dimensional datasets, the method remains effective.
> > >
> > >
> > >
> > > ## Causal-Aware Column Reweighting
> > >
> > > To further demonstrate the effectiveness of causal-aware column reweighting, we compare it with a supervised weight-learning baseline. Specifically, in our scoring framework, the final anomaly score for each sample can be viewed as being obtained by applying a fixed linear weighting to the vector of column-wise negative log-likelihood scores. Based on this formulation, we construct a learnable baseline with matching input and output behavior: we replace the fixed causal-aware reweighting with a trainable MLP, and train only the newly added module on a labeled dataset containing both normal and anomalous samples, while freezing the parameters of the original detector. We use the widely adopted weighted binary cross-entropy loss for supervised anomaly detection tasks. The corresponding AUC-ROC results on the fake job posts dataset are as follows:
> > >
> > > | Method | AUC-ROC |
> > > | --- | ---: |
> > > | AnoLLM (no reweighting) | 0.800 |
> > > | CausalTAD (causal-aware reweighting) | 0.873 |
> > > | 1-layer MLP | 0.881 |
> > > | 2-layer MLP | 0.890 |
> > > | 4-layer MLP | 0.889 |
> > > | 8-layer MLP | 0.891 |
> > >
> > > Conclusion: our **causal-aware column reweighting is already close to the best performance achieved by a supervised learnable head**. It does not rely on anomaly labels, yet it yields gains comparable to supervised learning.
> > >
> > >
> > >
> > >
> > > ## Statistical Significance Testing
> > >
> > > We have added results with 5 random seeds on the main datasets:
> > >
> > > | Dataset | AUC-ROC (mean ± std) |
> > > | --- | ---: |
> > > | Fake job posts | 0.871 ± 0.0035 |
> > > | Fraud ecommerce | 1.000 ± 0.0000 |
> > > | Lymphography | 0.998 ± 0.0032 |
> > > | Seismic | 0.783 ± 0.0057 |
> > > | Vehicle insurance | 0.580 ± 0.0057 |
> > >
> > > In the next version, we will complete more statistical significance results for all datasets. Meanwhile, for the parts of the paper that cite existing baseline results, we will make every effort to reproduce them and replace them where possible.

---

### Official Review · Reviewer_ud62 · 2026-03-13

**Soundness:** 2
**Presentation:** 3
**Significance:** 2
**Originality:** 2
**Overall Recommendation:** 3
**Confidence:** 3

**Summary:**

While recent a state-of-the-art tabular anomaly detection method based on LLM, AnoLLM, demonstrate strong performance, it fails to account for causal relationships between columns during serialization, potentially limiting detection accuracy. To address this gap, this work proposes CausalTAD, a novel framework comprising two key components: a causal-driven column ordering module and a causal-aware reweighting module. Empirical evaluation demonstrates the effectiveness of the proposed method.

**Compliance With Llm Reviewing Policy:**

Affirmed.

**Final Justification:**

I have carefully reviewed the authors’ response. Overall, the authors have actively addressed my concerns and provided clarifications that improve my understanding of the work.

However, considering the practical aspects of scalability and the incremental contribution, I maintain my original score.

**Key Questions For Authors:**

**Questions:**

1.	The experimental results show that increasing model size does not yield significant performance gains. Could the authors explain this phenomenon and discuss its implications for scaling the proposed approach?
2.	Regarding the column ordering problem during serialization, a simpler alternative would be to randomly permute columns and aggregate detection results via voting. Is such a strategy feasible, and how would its computational efficiency and detection performance compare to the proposed causal ordering module?

**Limitations:**

See Weaknesses

**Strengths And Weaknesses:**

**Strengths:**

1.	The paper is well-written and presents a clear research motivation.
2.	The proposed method offers a practical and feasible solution to a recognized limitation in existing work.
3.	The empirical evaluation is comprehensive and consistently demonstrates the effectiveness of the proposed method across multiple benchmarks.

**Weaknesses:**

1.	The contribution is incremental in nature, building upon existing LLM-based AD frameworks without introducing fundamentally new insights or methodologies.
2.	For high-dimensional tabular data, the causal-driven column ordering module incurs substantial computational cost, raising concerns about scalability to larger datasets and real-world deployment scenarios.

---

> ### Author Rebuttal · Authors · 2026-03-31
>
> # Response to Reviewer ud62
>
> Dear Reviewer ud62,
> Thank you for your constructive feedback.
> In response to the request for additional comparison against the latest baselines, we have added a comprehensive comparison with four recent strong baselines; please refer to the "Additional Baselines" section for details.
> We have also provided detailed responses to all of the Weaknesses, Key Questions, and Limitations you raised.
>
>
> # For Weaknesses
>
>
> ## The contribution is incremental in nature, building upon existing LLM-based AD frameworks without introducing fundamentally new insights or methodologies.
>
> Rather than a minor tweak to existing LLM-based anomaly detection frameworks, our method introduces a causal perspective by formulating causal-driven column ordering as a Linear Ordering Problem and applying causal-aware reweighting, thereby unifying causal structure, column ordering, and column reweighting into a robust tabular anomaly detection approach that delivers consistent gains on both mixed-type and purely numerical datasets.
>
>
>
> ## For high-dimensional tabular data, the causal-driven column ordering module incurs substantial computational cost
>
> We understand this concern; however, the main computation in our method is an **offline, dataset-level preprocessing step**, not part of model training or online inference. Specifically, factor extraction and column ordering are performed **once per dataset** and then reused, so they do not need to be repeated after preprocessing. This stage uses **no GPU resources** and can run in parallel with language-model fine-tuning across datasets, reducing overall time cost. Moreover, because our method does not modify the model architecture, it introduces **no additional inference-time overhead** for deployment.
>
>
> # For Key Questions
>
> ## increasing model size does not yield significant performance gains
>
> Experimental results show that scaling from 135M to 360M to 1.7B brings only marginal gains, and can even hurt performance on ODDS datasets dominated by numerical features. This indicates that the main bottleneck is not model capacity but task characteristics: tabular anomaly detection contains many numerical/categorical patterns with limited semantic redundancy, while over 98.5% of ODDS columns are numerical or categorical. Since large language models are pretrained mainly on text, they are less sensitive to dense numerical/categorical signals, reflecting a pronounced distribution shift. Therefore, for anomaly detection on mostly numerical/categorical datasets, smaller models can deliver lower cost and better accuracy; for future datasets with richer text content, larger models may offer more potential, and parameter scaling could be a reasonable first direction.
>
>
> ## randomly permute columns and aggregate detection results via voting
>
>
> Thank you for the suggestion. In the AnoLLM baseline paper, the authors also used random column ordering with voting-based aggregation, but the gain is highly sensitive to the number of test-time permutations. As reported in Appendix Fig. 3 (Sec. C), increasing permutations from 1 to 21 raises mean AUC-ROC only from 0.872 to 0.882, while variance decreases.
>
> However, this strategy substantially increases inference time: it requires repeated forward passes over multiple permutations, so inference cost grows linearly with the number of permutations (also discussed in Appendix Fig. 5, Sec. F). Therefore, we use a causal-driven column ordering module, where causal discovery is performed once offline, avoiding the multi-fold inference overhead of multi-permutation voting. This preserves detection quality and improves deployment practicality.
>
> Compared with random ordering plus voting, our main results (Sec. 5.2) show that our method consistently outperforms the AnoLLM baseline on more than 30 ODDS datasets.
>
>
> # Added Baselines
>
> ## comparison on datasets with free-text columns
> Comparisons on the **fakejob** dataset. The key metric, **AUC-ROC**, is as follows:
>
> | Method | AUC-ROC |
> |---|---:|
> | MCM | 0.676 |
> | DRL | 0.611 |
> | NPT-AD | 0.593 |
> | LLM-DAS (DRL) | 0.637 |
> | AnoLLM | 0.814 |
> | LLM-DAS (AnoLLM) | 0.846 |
> | CausalTAD-135M | **0.873** |
>
> **Conclusion:**
> - In the "**mixed-type datasets with free-text columns**" scenario we focus on, CausalTAD significantly outperforms the other recent baselines
>
> ## Comparison on datasets purely numerical and categorical datasets
> Results are taken directly from the original paper on the same datasets as ours(including BreastW, Wine, Arrhythmia, Mammography, Optdigits, Pendigits, Satimage-2, Shuttle, Thyroid, WBC, Satellite).
>
> | Dataset | DRL | MCM | LLM-DAS (PCA) | LLM-DAS (DRL) | NPT-AD | CausalTAD-135M |
> |---|---:|---:|---:|---:|---:|---:|
> | average | 0.957 | 0.946 | 0.732 | 0.958 | 0.941 | 0.942 |
>
> Although CausalTAD is indeed weaker than some recent baselines on certain datasets, the average gap is less than 0.02.

---

> > ### Author Rebuttal · Reviewer_ud62 · 2026-04-02
> >
> > Thanks for the authors' response.
> >
> > The concerns regarding the computational complexity and scalability of the casual ordering module have not been sufficiently addressed with detailed analysis and empirical evaluation. Furthermore, the discssion on the relationship between detection performance and model size remains unconvincing, particularly the claim that "large language models are pretrained mainly on text, they are less sensitive to dense numerical/categorical signals".
> >
> >  As a result, I maintain my original score.

---

> > > ### Author Response · Authors · 2026-04-06
> > >
> > > ## For High-Dimensional Tabular Datasets
> > >
> > > ### Time Cost
> > >
> > > The time cost is concentrated in two parts: 1. causal discovery and 2. column ordering.
> > >
> > > **1. Causal Discovery**
> > >
> > > The causal discovery stage requires one model call per sample, with the total input and output length under 4k. For GPT-4o, this takes 2 to 4 seconds (source: https://artificialanalysis.ai/models/gpt-4o). We conservatively use 5 seconds for estimation and account for API concurrency.
> > >
> > > If the dataset contains $n$ samples and the parallel concurrency level is $p$, then the theoretical time is:
> > >
> > > $$
> > > T(n,p)=\left\lceil \frac{n}{p}\right\rceil \cdot t_s,\qquad t_s=5\text{ s}
> > > $$
> > >
> > > For the largest-dimensional dataset in the ODDS library, speech: $n=3686$, $m=400$, the theoretical runtime under different concurrency levels is:
> > >
> > > | Concurrency $p$ | Number of batches $\lceil 3686/p\rceil$ | Theoretical runtime |
> > > |---|---:|---:|
> > > | 1 | 3686 |5 hours 7 minutes |
> > > | 4 | 922 | 1 hour 16 minutes |
> > > | 8 | 461 | 38 minutes |
> > > | 16 | 231 | 19 minutes |
> > > | 32 | 116 | 9 minutes |
> > >
> > > **2. column ordering**
> > >
> > > The core issue is the Linear Ordering Problem (LOP). Since it is NP-hard, a polynomial-time exact worst-case upper bound is not expected.
> > >
> > > Our method does not aim to find the absolute optimal ordering in every case, but rather a sufficiently good ordering within a fixed time budget. On the 400-column Speech dataset, we evaluated different search budgets:
> > >
> > > | Time Limit | AUC-ROC |
> > > |---|---:|
> > > | AnoLLM baseline | 0.470 |
> > > | 60 s | 0.593 |
> > > | 300 s | 0.601 |
> > > | 600 s | 0.621 |
> > > | 1200 s | 0.623 |
> > > | 2400 s | 0.622 |
> > > | 3600 s | 0.624 |
> > > | Full search | 0.625 |
> > >
> > > A budget of around 600 seconds is already enough to reach performance close to full search.
> > >
> > > **3. Conclusion**
> > >
> > > Even for a 400-dimensional dataset, the method can finish in about **50 minutes** total. In an enterprise setting, for example with DeepSeek-R1-Distill-Qwen-32B and higher concurrency, the total time cost can be reduced to **within 20 minutes**.
> > >
> > >
> > > ### Performance
> > >
> > > We further validated the method on more high-dimensional datasets. The AUC-ROC results are:
> > >
> > > | Dataset | Number of Columns | CausalTAD | AnoLLM |
> > > | --- | ---: | ---: | ---: |
> > > | Speech | 400 | 0.625 | 0.470 |
> > > | Musk | 166 | 1.000 | 1.000 |
> > > | Arrhythmia | 274 | 0.836 | 0.803 |
> > >
> > > On high-dimensional datasets, the method remains effective.
> > >
> > >
> > >
> > > ## On the Relationship Between Detection Performance and LLM backbone size
> > >
> > > 1. In our fine-tuning setting, a more reasonable interpretation is not that “larger models are always better,” but that once the LLM backbone already has sufficient expressive capacity, the task itself becomes the bottleneck for useful signal. A JMIR study shows that task-specific fine-tuning often exhibits clear diminishing returns, and that training data quality and the intended use of the model may be equally or even more important than LLM backbone size [Majdik et al., 2024](https://ai.jmir.org/2024/1/e52095/). Therefore, **when performing anomaly detection on limited tabular samples, further scaling the LLM backbone often yields only marginal gains** and does not lead to stable or significant improvements.
> > >
> > > 2. For tabular anomaly detection, the key constraint lies more in structural representation than in raw capacity. Table Meets LLM finds that LLM performance on structured table tasks is highly sensitive to table input format, content order, role prompting, and partition marks, indicating that the organization of the input itself is an important determinant of performance [Sui et al., 2024](https://arxiv.org/abs/2305.13062). Consistently, the large-scale benchmark in TALENT shows that heterogeneous interactions between categorical and numerical attributes often determine which class of methods performs better, and that the best performance is concentrated in only a few models rather than increasing monotonically with LLM backbone size [Ye et al., 2024](https://arxiv.org/abs/2407.00956). This is highly consistent with our setting: our data is mainly composed of numerical and categorical fields, with only a small amount of free-text columns, so the main factor limiting detection performance is not LLM backbone size, but whether the model can preserve and exploit dependencies among fields.
> > >
> > > 3. Therefore, our results should be understood as a bounded regime phenomenon that is partly affected by random fluctuations: across different LLM backbones such as 135M, 360M, and 1.7B, the AUC only varies slightly within 0.865–0.876, indicating that under the current setting, increasing parameters does not open up a new performance regime. For future extension, a more effective direction is not simply to enlarge the LLM backbone, but to expand richer tabular samples, especially mixed-type datasets with stronger textual signals, and to further improve serialization and causal alignment. This is precisely the significance of CausalTAD: it provides a structural enhancement, rather than a method that only works by relying on larger model capacity.

---

### Official Review · Reviewer_B42e · 2026-03-13

**Soundness:** 2
**Presentation:** 2
**Significance:** 3
**Originality:** 2
**Overall Recommendation:** 4
**Confidence:** 4

**Summary:**

The current paper proposes to introduce causal knowledge to the serialization of tabular data to improve tabular anomaly detection. Classical LLM approaches such as AnoLLM convert tabular rows into textual sequences to train language models to predict anomaly detection probabilities. Typically, columns are treated in random order, therefore loosing correlated dependencies between features. Here, the CausalTAD method performs a causal-driven column ordering followed by a causal-aware reweighting method. A benchmark is performed on 30 datasets and show that CausalTAD consistently outperforms SOTA methods.

**Compliance With Llm Reviewing Policy:**

Affirmed.

**Final Justification:**

The rebuttal satisfactorily addressed my main concerns: added experiments on LLM robustness and the extended appendix about causal ordering and weights improve the clarity and reproducibility of the work.

While comparisons with alternative feature-ordering heuristics are limited and some concerns about originality and presentation persist, these weaknesses no longer outweigh the strengths.

I therefore update my recommendation to borderline accept (4).

**Key Questions For Authors:**

1.	How does this method compare with other feature-ordering heuristics?
2.	Since the method reweights columns based on causality, it would be interesting to include in the supplementary all weights and columns causality.
3.     How does the model and the methods are dependent on the LLM? A benchmark on different LLMs would clarify that point.

**Limitations:**

The authors include an Impact Statement acknowledging that anomaly detection systems may be misused or over-relied upon and that human oversight is required.

**Strengths And Weaknesses:**

Interesting idea bridging a gap between column causality and LLM-based anomaly detection. The conception is clear and the benchmark is performed on an important number of different datasets. Figures captions are too short while they should be the more self contained possible and panels should be numbered and explained in the text and the caption. The overall paper looks more like a combination of techniques than a novel ML architecture in that field. The introduction is slithly too much is not always supported by references. The Figure 1 should not be explained in the introduction, but later one when introducing the approach and with less details.

---

> ### Author Rebuttal · Authors · 2026-03-31
>
> # Response to Reviewer B42e
>
> Dear Reviewer B42e,
> Thank you for your constructive feedback.
> In response to the request for additional recent baselines, we have added a comprehensive comparison against four recent strong baselines; please see the "Additional Baselines" section for details.
>
>
>
> ## How does this method compare with other feature-ordering heuristics?
>
> Thank you for the suggestion. After searching the accepted literature, we found that **mature feature-ordering schemes for tabular data are quite rare**. In particular, in the anomaly detection setting, the mainstream consensus is usually that tabular data are approximately permutation-invariant with respect to column order. Therefore, it is currently difficult to find a mature ordering baseline that can be **directly compared** with our method.
> For why we still attempted a column-ordering method despite this consensus in the field, please refer to our response to reviewer ik6q regarding the question of "the core assumption of the paper."
> Meanwhile, in the scenario of "contamination-free unsupervised tabular anomaly detection with free-text columns" that we target, constructing other comparable ordering methods would require handling numerical / categorical / textual multimodal features simultaneously while strictly satisfying the unsupervised setting, which makes the implementation rather complex. Due to rebuttal time constraints, we are unable to complete this additional experiment at this stage. In a future version, we will add and systematically compare more ordering strategies (such as MI-based and Variance/Entropy-based ones).
>
>
>
> ## Since the method reweights columns based on causality, it would be interesting to include in the supplementary all weights and columns causality.
>
> Thank you for pointing this out. We have already expanded the appendix, which will include the causal-driven column ordering, the weights, and the original factor-level causal graph on all of the major datasets.
>
>
>
> ## How does the model and the methods are dependent on the LLM?
>
> **Impact of different LLMs for casual discovery**
> We evaluated our framework with different language models on the **fakejob** dataset, and the resulting **AUC-ROC** scores are shown below:
>
> | Language Model | AUC-ROC |
> |---|---:|
> | GPT-4 | 0.873 |
> | GPT-5.4 | 0.872 |
> | GPT-5 mini | 0.867 |
> | DeepSeek-R1-Distill-Qwen-32B (self-deployed) | 0.869 |
>
> These results show that our method is **highly consistent across different types of language models**. Even with the self-deployed DeepSeek-R1-Distill-Qwen-32B, which we ran under limited computational resources, the method still maintains strong performance.
>
>
> **Impact of different LLMs for finetuning**
> To evaluate the sensitivity of our method to the fine-tuned base model, we conducted an additional experiment on the fakejob dataset.
> The newly added fine-tunable LLMs are SmolLM2-135M and the Qwen3 family (0.6B and 1.7B).
>
> | Model | AUC-ROC |
> |---|---:|
> | SmolLM-135M | 0.873 |
> | SmolLM-360M | 0.868 |
> | SmolLM-1.7B | 0.876 |
> | SmolLM2-135M | 0.868 |
> | Qwen3-0.6B | 0.865 |
> | Qwen3-1.7B | 0.875 |
>
> The performance across different models is overall very close, with a maximum difference of **0.011**, indicating that the proposed method is reasonably stable and transferable across different fine-tuned LLMs.
>
>
>
> # Added Baselines
>
> ## comparison on datasets with free-text columns
> Comparisons on the **fakejob** dataset. The key metric, **AUC-ROC**, is as follows:
>
>
> | Method | AUC-ROC |
> |---|---:|
> | MCM | 0.676 |
> | DRL | 0.611 |
> | NPT-AD | 0.593 |
> | LLM-DAS (DRL) | 0.637 |
> | AnoLLM | 0.814 |
> | LLM-DAS (AnoLLM) | 0.846 |
> | CausalTAD-135M | **0.873** |
>
> **Conclusion:**
> - In the "**mixed-type datasets with free-text columns**" scenario we focus on, CausalTAD significantly outperforms the other recent baselines
> - We tried using the LLM-DAS data augmentation method to enhance our AnoLLM. It still remains weaker than our CausalTAD
>
> ## Comparison on datasets purely numerical and categorical datasets
> Results are taken directly from the original paper on the same datasets as ours(including BreastW, Wine, Arrhythmia, Mammography, Optdigits, Pendigits, Satimage-2, Shuttle, Thyroid, WBC, Satellite).
>
> | Dataset | DRL | MCM | LLM-DAS (PCA) | LLM-DAS (DRL) | NPT-AD | CausalTAD-135M |
> |---|---:|---:|---:|---:|---:|---:|
> | average | 0.957 | 0.946 | 0.732 | 0.958 | 0.941 | 0.942 |
>
> Although CausalTAD is indeed weaker than some recent baselines on certain datasets, the average gap is less than 0.02.

---

> > ### Author Rebuttal · Reviewer_B42e · 2026-04-02
> >
> > The authors clarified the robustness of the method across different LLMs with experiments, and they expanded the appendix to include causal ordering and weights, which improves transparency and reproducibility. While comparisons with alternative feature-ordering heuristics are still limited, the authors provide a reasonable justification given the current state of the literature and acknowledge this as future work.

---

> > > ### Author Response · Authors · 2026-04-06
> > >
> > > Dear Reviewer B42e,
> > > Thank you for your acknowledgement and for the helpful feedback throughout the review process. We are pleased that the added robustness experiments across different LLMs, together with the appendix updates on causal ordering and weights, have resolved your concerns. We also appreciate your suggestion regarding alternative feature-ordering heuristics.

---

### Decision · Program_Chairs · 2026-04-30

**Decision:**

Reject

**Comment:**

The paper proposes CausalTAD, an LLM-based tabular anomaly detection framework that incorporates causal knowledge through causal-driven column ordering and causal-aware reweighting. Experiments on a broad collection of datasets show improvements over the main LLM-based baseline.

The reviewers agree that the paper addresses an important problem and that the proposed causal perspective is interesting and practically relevant. They also find the empirical study fairly broad, and the rebuttal helped clarify several aspects of the method by adding stronger baselines, LLM-sensitivity analysis, high-dimensional experiments, and more discussion of the causal discovery pipeline. However, concerns remain about the incremental nature of the contribution, the heuristic character of some key design choices, scalability, and the strength and completeness of the evaluation relative to recent baselines. Overall, while the paper has merit, it is not sufficiently strong for acceptance at ICML in its current form.